# Enhancing Multi-Modal LLMs Reasoning via Difficulty-Aware Group Normalization

## Abstract

Reinforcement Learning with Verifiable Rewards (RLVR) and Group Relative Policy Optimization (GRPO) have significantly advanced the reasoning capabilities of large language models. Extending these methods to multimodal settings, however, faces a critical challenge: the instability of *std*-based normalization, which is easily distorted by extreme samples with nearly positive or negative rewards. Unlike pure-text LLMs, multimodal models are particularly sensitive to such distortions, as both perceptual and reasoning errors influence their responses. To address this, we characterize each sample by its **difficulty**, defined through perceptual complexity (measured via visual entropy) and reasoning uncertainty (captured by model confidence). Building on this characterization, we propose **difficulty-aware group normalization**, which re-groups samples by difficulty levels and shares the *std* within each group. Our approach preserves GRPO's intra-group distinctions while eliminating sensitivity to extreme cases, yielding significant performance gains across multiple multimodal reasoning benchmarks.

## 1 Introduction

Reinforcement Learning with Verifiable Rewards (RLVR) has enabled significant advances in the reasoning capabilities of both large language models (LLMs) (DeepSeek-AI et al., 2025; Yang et al., 2025a; Lambert et al., 2024) and multi-modal large language models (MLLMs) (Zhang et al., 2025b; Huang et al., 2025a). Within this paradigm, Group Relative Policy Optimization (GRPO) (Shao et al., 2024b) demonstrates strong performance by applying standard deviation (*std*)-based normalization to rewards within each response group. This *std*-based normalization rescales intra-group distinctions between positive and negative responses, thereby stabilizing training.

Despite these advances, the *std*-based normalization suffers from a critical limitation: *sensitive to extreme samples* — those with response groups that are almost entirely positive or negative. Specifically, when rewards in a group collapse to near 0 or 1, the resulting low *std* overemphasizes the extreme samples during optimization. Meanwhile, samples with more balanced rewards are neglected, leading to imbalanced optimization. This issue is particularly pronounced in MLLMs, where the complexity of multimodal inputs increases the occurrence of such extreme samples. As illustrated in Figure 1, MLLM responses are jointly influenced by challenges from perceptual complexity and reasoning uncertainty, making them more susceptible to extreme reward distributions.

While removing the *std* term mitigates the risk of overfitting to extreme samples (Liu et al., 2025b), it simultaneously discards the valuable intra-group distinctions, which are essential for effective and stable optimization. Therefore, *the key issue lies not in the std-normalization itself, but rather in the way groups are constructed*: when group size is small, extreme cases become inevitable. Enlarging group sizes during rollouts could help, but it incurs prohibitive computational costs.

Motivated by this, we propose to account for the challenges of various samples, which we refer to as **difficulty-aware re-grouping**. We characterize each sample's difficulty from two complementary perspectives: (i) a data-centric view, where high entropy in the image reflects *perceptual difficulty*; and (ii) a model-centric view, where low confidence in model responses reflects *reasoning difficulty*. By re-grouping samples according to these difficulty levels and sharing the *std* within each group, our method preserves intra-group distinctions while mitigating sensitivity to extreme cases. Specifically, our difficulty-based re-group strategy is achieved by:

**Perceptual difficulty based regrouping.** We quantify perceptual difficulty through spectral analysis of image patch covariances, where higher entropy in the resulting eigenvalue distribution indicates greater visual complexity. Intuitively, images with more diverse and complex visual patterns exhibit higher entropy, reflecting greater perceptual difficulty.

**Reasoning difficulty based regrouping.** Leveraging the insight that token-level log probabilities reflect reasoning confidence (Yu et al., 2025c), we measure reasoning difficulty through the model's token-level confidence, where lower average log probabilities indicate greater uncertainty in generating correct reasoning chains, reflecting higher reasoning difficulty.

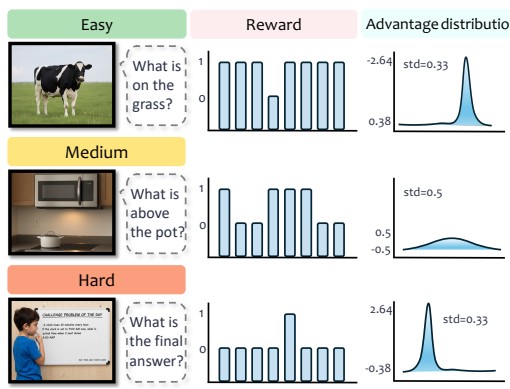

Figure 1: Advantage distribution after the normalization of reward varies among samples. Extreme samples like easy and hard ones are amplified after *std*-normalization, whereas medium samples exhibit more balanced advantages.

By explicitly decomposing difficulty into data-centric (**perceptual**) and model-centric (**reasoning**) groups, our method allows each group of samples to share separate *std*s for perceptual and reasoning aspects. These normalized advantages are then combined via element-wise multiplication, effectively integrating intrinsic data complexity and model uncertainty, ensuring stable optimization that preserves meaningful intra-group distinctions. Our contributions are summarized into the following three aspects:

- We identify that the limitation of *std*-normalization in group-based RL methods (*e.g.* GRPO and DAPO) originates from group construction, particularly in multimodal reasoning tasks, and propose a difficulty-based re-grouping strategy to build more robust groups.
- We explicitly decompose difficulty into perceptual and reasoning aspects, and integrate them via element-wise combination, effectively capturing both data complexity and model uncertainty while preserving intra-group distinctions.
- Our proposed strategy achieves significant performance gains. Concretely, building upon GRPO and DAPO, our strategy attains more than 2% average performance improvements.

## 2 PRELIMINARY

In this section, we briefly introduce the key concepts and training setup for multimodal reasoning under RLVR (DeepSeek-AI et al., 2025). We first formulate the task, and then revisit the standard GRPO framework (Shao et al., 2024b) along with its improved variant, Decoupled Clip and Dynamic Sampling Policy Optimization (DAPO) (Yu et al., 2025b).

### 2.1 TASK FORMULATION

We consider the problem of multimodal reasoning under the RLVR paradigm. Let $\{\mathcal{I}, \mathcal{Q}\} \in \mathcal{D}$ denote a multimodal input, where the dataset $\mathcal{D}$ includes image $\mathcal{I}$ and text question $\mathcal{Q}$. The model generates a reasoning response $o$ given $\{\mathcal{I}, \mathcal{Q}\}$ and receives a verifiable reward $r$ based on the correct answer $y$ (Wu et al., 2025; Wang et al., 2025a). The response $o$ typically contains both the reasoning steps and the final answer, with the reasoning steps enclosed in `<think>...</think>` and the final answer enclosed in `\boxed{}`. We employ a binary reward function, where $r(o, y) = 1$ if the final answer is equal to the correct answer $y$, and $r(o, y) = 0$ otherwise. The reasoning process is modeled as a policy $\pi_\theta(o|\mathcal{I}, \mathcal{Q})$ parameterized by $\theta$ to maximize the expected reward:

$$\mathcal{J}_{\text{RLVR}}(\theta) = \max_\theta \mathbb{E}_{\{\mathcal{I}, \mathcal{Q}\} \sim \mathcal{D}} \mathbb{E}_{o \sim \pi_\theta(\cdot|\mathcal{I}, \mathcal{Q})}[r(o, y)]. \tag{1}$$

Our goal is to enhance the reasoning capabilities of an instruction-tuned MLLM, thereby significantly improving its performance on downstream multimodal reasoning tasks.

## 2.2 CORE ALGORITHMS OF REINFORCEMENT LEARNING WITH VERIFIABLE REWARD

**Group Relative Policy Optimization (GRPO)** is derived from Proximal Policy Optimization (PPO) (Schulman et al., 2017), with the key distinction that GRPO replaces the advantage estimates obtained via Generalized Advantage Estimation (GAE) with group-relative advantages computed from a group of outputs.

Specifically, for each input $\mathcal{I}, \mathcal{Q}$, GRPO samples a group of outputs $\{o_1, o_2, \ldots, o_G\}$ from the old policy model $\pi_{\theta_{\text{old}}}$, with rollout size $G$. The advantage of the $i$-th response is computed by normalizing the rewards among the group:

$$\hat{A}_i = \frac{r_i - mean(\{r_1, r_2, \ldots, r_G\})}{std(\{r_1, r_2, \ldots, r_G\})}. \tag{2}$$

GRPO adopts a clipped objective, together with a directly imposed KL penalty term:

$$\mathcal{J}_{\text{GRPO}}(\theta) = \mathbb{E}_{(\mathcal{I},\mathcal{Q})\sim\mathcal{D},\{o_i\}_{i=1}^G \sim \pi_{\theta_{\text{old}}}(o|\mathcal{I},\mathcal{Q})} \left\{ \frac{1}{G} \sum_{i=1}^G \frac{1}{|o_i|} \sum_{t=1}^{|o_i|} \right. \tag{3}$$

$$\min \left[ \frac{\pi_\theta\left(o_{i,t} \mid \mathcal{I}, \mathcal{Q}, o_{i,<t}\right)}{\pi_{\theta_{\text{old}}}\left(o_{i,t} \mid \mathcal{I}, \mathcal{Q}, o_{i,<t}\right)} \hat{A}_{i,t}, \ \text{clip}\left(\frac{\pi_\theta\left(o_{i,t} \mid \mathcal{I}, \mathcal{Q}, o_{i,<t}\right)}{\pi_{\theta_{\text{old}}}\left(o_{i,t} \mid \mathcal{I}, \mathcal{Q}, o_{i,<t}\right)}, 1-\epsilon, 1+\epsilon\right) \hat{A}_{i,t} \right] - \beta \mathbb{D}_{KL}(\pi_\theta \| \pi_{\text{ref}}) \right\}.$$

$\epsilon$ is the hyperparameter to control the clipping range of the importance sampling ratio, and $\beta$ is the penalty strength of how far the current policy $\pi_\theta$ deviates from the reference policy $\pi_{ref}$.

**Decoupled Clip and Dynamic Sampling Policy Optimization (DAPO)** is a variant of GRPO adopting an asymmetric clipping range with a larger upper bound, dynamic sampling, token-level policy gradient loss, and overlong reward shaping. The objective function of DAPO is defined as:

$$\mathcal{J}_{\text{DAPO}}(\theta) = \mathbb{E}_{(\mathcal{I},\mathcal{Q})\sim\mathcal{D}, \{o_i\}_{i=1}^G \sim \pi_{\theta_{\text{old}}}(o|\mathcal{I},\mathcal{Q})} \left\{ \frac{1}{\sum_{i=1}^G |o_i|} \sum_{i=1}^G \sum_{t=1}^{|o_i|} \right. \tag{4}$$

$$\min \left[ \frac{\pi_\theta\left(o_{i,t} \mid \mathcal{I}, \mathcal{Q}, o_{i,<t}\right)}{\pi_{\theta_{\text{old}}}\left(o_{i,t} \mid \mathcal{I}, \mathcal{Q}, o_{i,<t}\right)} \hat{A}_{i,t}, \ \text{clip}\left(\frac{\pi_\theta\left(o_{i,t} \mid \mathcal{I}, \mathcal{Q}, o_{i,<t}\right)}{\pi_{\theta_{\text{old}}}\left(o_{i,t} \mid \mathcal{I}, \mathcal{Q}, o_{i,<t}\right)}, 1-\epsilon_{\text{low}}, 1+\epsilon_{\text{high}}\right) \hat{A}_{i,t} \right] \right\}.$$

## 3 METHOD

In this section, we introduce our difficulty-based regroup strategy in detail. We first represent our perceptual difficulty-based regrouping in Section 3.1, then we describe our reasoning difficulty-based regrouping in Section 3.2. Finally, we show our combination strategies.

### 3.1 PERCEPTUAL DIFFICULTY-BASED RE-GRPOUPING

**Perceptual difficulty estimation.** To estimate the perceptual difficulty of a batch $\mathcal{B} = \{(\mathcal{I}_s, \mathcal{Q}_s)\}_{s=1}^B$, we first extract patch-level visual features from the Qwen2.5-VL-7B visual encoder $\mathbf{\Omega}_v$:

$$\mathbf{F}_s = \mathbf{\Omega}_v(\mathcal{I}_s) \in \mathbb{R}^{P \times d} = [\boldsymbol{f}_s^1, \boldsymbol{f}_s^2, \cdots, \boldsymbol{f}_s^P]^\top, \tag{5}$$

where $P$ denotes the number of spatial patches and $d$ is the feature dimension, with $\boldsymbol{f}_s^j \in \mathbb{R}^{d \times 1}, j = 1, \cdots, P$ representing the feature of the $j$-th patch.

Compared to CLIP-based representations (Radford et al., 2021), these patch-level features not only capture finer spatial granularity that preserves local details, but also align better with the downstream textual decoder $\mathbf{\Omega}_t$, ensuring both stability and semantic consistency.

We then compute the empirical covariance matrix to capture intra- and inter-patch variances:

$$\mathbf{C}_s = \frac{1}{P-1}(\mathbf{F}_s - \mathbf{1}_P \boldsymbol{\mu}_s^\top)(\mathbf{F}_s - \mathbf{1}_P \boldsymbol{\mu}_s^\top)^\top, \quad \boldsymbol{\mu}_s = \frac{1}{P}\sum_{j=1}^P \boldsymbol{f}_s^j. \tag{6}$$

where $\mathbf{1}_P$ is a $P \times 1$ column vector of ones and $\boldsymbol{\mu}_s$ is the mean of the patches feature of the $\mathcal{I}_s$. The diagonal entries measure the variance of each feature dimension across patches, while the off-diagonal

terms capture correlations between different feature dimensions. This covariance structure reveals whether visual features are dominated by a few strong dimensions or by multiple interacting factors, providing a principled basis for assessing perceptual difficulty.

Since $\mathbf{C}_s$ is a symmetric positive semidefinite matrix, we then perform eigenvalue decomposition for spectral analysis:

$$\mathbf{C}_s = \mathbf{V}_s \mathbf{\Lambda}_s \mathbf{V}_s^\top, \quad \mathbf{\Lambda}_s = \mathrm{diag}(\lambda_s^1, \ldots, \lambda_s^P), \ \lambda_s^k \geq 0. \tag{7}$$

$\lambda_s^k$ denotes the $k$-th eigenvalue, quantifying the variance along one orthogonal principal direction. Concentrated eigenvalues indicate that most variance is captured by a few dimensions, whereas more balanced eigenvalues imply richer visual structure and higher visual complexity.

The final perceptual difficulty score is defined as the entropy (Shannon, 1948) of the normalized distribution of eigenvalues:

$$H(\mathcal{I}_s) = -\sum_{k=1}^{P} p_s^k \log p_s^k, \tag{8}$$

where $p_s$ is the normalized probability distribution of the eigenvalues, and each element in $p_s$ is calculated as:

$$p_s^k = \frac{\lambda_s^k}{\sum_{j=1}^{P} \lambda_s^j}, \quad \text{with} \sum_{k=1}^{P} p_s^k = 1. \tag{9}$$

Here, low entropy corresponds to the visually easy sample, with variance concentrated on a few dominant components, whereas high entropy indicates the visually difficult sample, with variance distributed across many patches.

**Perceptual difficulty-based regrouping.** Given the perceptual difficulty scores within a batch, we partition samples into three groups using the 25th and 75th percentiles $\tau_{0.25}$ and $\tau_{0.75}$:

$$\mathcal{S}_1 = \{s \mid H(\mathcal{I}_s) \leq \tau_{0.25}\}, \quad \mathcal{S}_2 = \{s \mid \tau_{0.25} < H(\mathcal{I}_s) < \tau_{0.75}\}, \quad \mathcal{S}_3 = \{s \mid H(\mathcal{I}_s) \geq \tau_{0.75}\}. \tag{10}$$

For each group $a$, the reward set can be defined as:

$$\mathcal{R}_a = \{r_{s,i} \mid i = 1, \cdots, G, s \in \mathcal{S}_a\}, \ a \in \{1, 2, 3\} \tag{11}$$

where $r_{s,i}$ refers to the $i$-th reward of the $s$-th sample which belongs to the group $\mathcal{S}_a$.

We then compute the shared standard deviation $std(\mathcal{R}_a)$ of group rewards, and normalize the reward of each sample in batch with the new $std(\mathcal{R}_a)$ to calculate advantage accordingly:

$$A_{s,i}^{\text{Perceptual}} = \frac{r_{s,i} - mean(r_{s,1}, r_{s,2}, \ldots r_{s,G})}{std(\mathcal{R}_a)}, \tag{12}$$

$$std(\mathcal{R}_a) = \sqrt{\frac{1}{|\mathcal{R}_a| - 1} \sum_{r_{s,i} \in \mathcal{R}_a} \left(r_{s,i} - \frac{1}{|\mathcal{R}_a|} \sum_{r_{s,i} \in \mathcal{R}_a} r_{s,i}\right)^2}, \tag{13}$$

where $|\mathcal{R}_a|$ denotes to the cardinality of $\mathcal{R}_a$.

By grouping samples into low-, medium-, and high-entropy categories, the normalization scale is shared only among samples with comparable perceptual difficulty. This mitigates the influence of extreme samples, balances treatment across different levels of visual complexity, and ultimately stabilizes optimization.

## 3.2 REASONING DIFFICULTY-BASED REGROUPING

**Reasoning difficulty estimation.** While perceptual difficulty captures the intrinsic complexity of the image, reasoning difficulty is shaped by the model's intrinsic confidence in generating the final answer. Even for inputs with similar visual complexity, the model may exhibit varying confidence levels: high confidence (assigning a high probability to reasoning chains) implies a clear and reliable reasoning path, whereas low confidence indicates uncertainty and potential reasoning failures. Following this intuition, we quantify reasoning difficulty using the model's probabilities to its reasoning chains.

For the given batch $\mathcal{B} = \{(\mathcal{I}_s, \mathcal{Q}_s)\}_{s=1}^{B}$, and the generated $G$ responses for each sample, we denote the $i$-th response as $o_{s,i} = (o_{s,i}^1, \ldots, o_{s,i}^T)$, where $o_{s,i}^n$ is the $n$-th token and $T$ is the sequence length.

Based on the token-level log probability $\pi_\theta \left( o_{s,i}^n \,\middle|\, \mathcal{I}_s, \mathcal{Q}_s, o_{s,i}^{<n} \right)$, we aggregate across tokens to obtain the sequence-level log probability for response $o_{s,i}$:

$$L_{s,i} = \sum_{n=1}^{T} \pi_\theta \left( o_{s,i}^n \,\middle|\, \mathcal{I}_s, \mathcal{Q}_s, o_{s,i}^{<n} \right). \tag{14}$$

Then we define the model confidence for sample $(\mathcal{I}_s, \mathcal{Q}_s)$ as the average sequence-level log probability across its $G$ rollouts:

$$L(\mathcal{Q}_s) \;=\; \frac{1}{G} \sum_{i=1}^{G} L_{s,i}. \tag{15}$$

This formulation reflects the model's internal confidence: High and consistent $L(\mathcal{Q}_s)$ indicates a reliable reasoning chain, whereas low or fluctuating $L(\mathcal{Q}_s)$ reflects epistemic uncertainty, implying that more challenging reasoning sample.

**Reasoning difficulty-based regrouping.** Given the model confidence scores $L(\mathcal{Q}_s)$ for the batch $\mathcal{B} = \{(\mathcal{I}_s, \mathcal{Q}_s)\}_{s=1}^{B}$, we divide samples into $b$ groups according to the quantiles of their confidence distribution. Let $\{\tau_0, \tau_1, \ldots, \tau_b\}$ denote the quantile boundaries, with $\tau_0 = 0$ and $\tau_b = 1$. Each question $\mathcal{Q}_s$ is then assigned to a group by:

$$\mathcal{M}_u = \{ s \mid \tau_{u-1} \le L(\mathcal{Q}_s) < \tau_u \}, \quad u \in \{1, \ldots, b\}. \tag{16}$$

Within each group $\mathcal{M}_u$, we define the reward set as:

$$\mathcal{R}_u = \{r_{s,i} \mid i = 1, \ldots, G, s \in \mathcal{M}_u\}, \; u \in \{1, \ldots, b\} \tag{17}$$

where $r_{u,i}$ is the reward of the $i$-th response for the sample, which belongs to the $u$ group. We can then calculate the shared standard deviation $std(\mathcal{R}_\mathcal{W})$ of reasoning difficulty-based group, and compute the advantage accordingly:

$$A_{s,i}^{\text{Reasoning}} = \frac{r_{s,i} - mean(r_{s,1}, r_{s,2}, \ldots r_{s,G})}{std(\mathcal{R}_u)}, \tag{18}$$

where the *std* can be calculated as:

$$std(\mathcal{R}_u) = \sqrt{\frac{1}{|\mathcal{R}_u| - 1} \sum_{r_{s,i} \in \mathcal{R}_u} \left( r_{s,i} - \frac{1}{|\mathcal{R}_u|} \sum_{r_{s,i} \in \mathcal{R}_u} r_{s,i} \right)^2}, \tag{19}$$

This regrouping ensures that responses with similar confidence levels are normalized on comparable scales, mitigating instability introduced by overconfident or underconfident samples.

## 3.3 COMBINATION FOR ROBUST OPTIMIZATION

To leverage the complementary aspects of perceptual and reasoning difficulty, we propose an element-wise combination strategy. Specifically, given the perceptual-based group normalized advantage $A^{\text{Perceptual}}$, the reasoning-based group normalized advantage $A^{\text{Reasoning}}$, and the original GRPO advantage $A^{\text{GRPO}}$, the combined advantage is defined as:

$$A^{\text{Combined}} \;=\; \alpha_{\text{Ori}} \cdot A^{\text{GRPO}} \;+\; \alpha_{\text{Percep}} \cdot A^{\text{Perceptual}} \;+\; \alpha_{\text{Reason}} \cdot A^{\text{Reasoning}}, \tag{20}$$

where $\alpha_{\text{Ori}}, \alpha_{\text{Percep}}, \alpha_{\text{Reason}}$ are weighting coefficients that balance the contributions of the three components. Perceptual difficulty, quantified by the entropy in the image, captures the *visual complexity* of multimodal inputs; reasoning difficulty, derived from token- and sequence-level log probabilities, reflects the *model uncertainty* during reasoning. Integrating these difficulty-based advantages with the original GRPO advantage allows the model to preserve meaningful intra-sample distinctions and incorporate both intrinsic and extrinsic difficulty context, providing a more stable and informative advantage for policy optimization.

Table 1: The Acc performance of two re-grouping strategies in our method. We take DAPO as our backbone. The best results are indicated in **boldface**, and the second-best results are underlined.

| Model | MathVerse | MathVision | MathVista | WeMath | HallusionBench | Avg |
|---|---|---|---|---|---|---|
| DAPO | 50.4 | 27.6 | 70.7 | 69.4 | 68.6 | 57.3 |
| DAPO + Perceptual regrouping | **52.3** | 28.0 | 71.4 | 70.9 | 70.9 | 58.7 |
| DAPO + Reasoning regrouping | **52.3** | 28.3 | 71.6 | 70.8 | 68.9 | 58.4 |
| **DAPO + Both** | 51.9 | **29.0** | **72.2** | **71.8** | **71.4** | **59.3** |

## 4 EXPERIMENT

In this section, we demonstrate the effectiveness of our re-grouping strategy. Specifically, we first show the experimental settings, including the dataset, benchmark, baselines, and implementation details. Then, we illustrate the ablation studies to analyze the two regroup strategies. Finally, we compare with the state-of-the-art methods over various benchmarks.

### 4.1 EXPERIMENTAL SETTINGS

**Dataset.** For training, we rely on the Geometry3K (Lu et al., 2021) dataset, which provides 2.1K training samples and 0.3K validation samples. Besides, we also provide the experimental results training on a larger dataset ViRL39k.

**Benchmark.** We evaluate our method on five benchmarks: four visual reasoning datasets, namely MathVerse (Zhang et al., 2024), MathVision (Wang et al., 2024), MathVista (Lu et al., 2024), and WeMath (Qiao et al., 2025), as well as one visual perception benchmark, HallusionBench (Guan et al., 2024). In addition, we assess the in-domain performance by comparing our method with the vanilla GRPO and DAPO.

**Baseline.** To evaluate the performance of our method, we consider three categories of baselines: (1) Closed-source models, such as GPT-4o (Hurst et al., 2024), and Cloud-3.5-sonnet (Anthropic, 2024). (2) Open source models: including InternVL-2.5-8B-Instruct (Chen et al., 2024), LLaVA-OneVision-7B (Li et al., 2024), Kimi-VL-16B (Du et al., 2025a), URSA-8B (Luo et al., 2025), and Mulberry-7B (Yao et al., 2024). (3) RLVR-based Models: MLLMs trained with reinforcement learning using verifiable rewards, representing the current mainstream approaches in this line of research. This category includes R1-VL-7B (Zhang et al., 2025a) , Vision-R1-7B (Huang et al., 2025b) , R1-OneVision-7B (Yang et al., 2025b) , OpenVLThinker-7B (Deng et al., 2025) , MM-Eureka-Qwen-7B (Meng et al., 2025) , ADORA-7B (Gui & Ren, 2025) , ThinkLite-7B-VL (Wang et al., 2025d) , and VLAA-Thinker-7B (Chen et al., 2025a) .

**Implementation details.** Following prior work (Liu et al., 2025a), we use Qwen2.5-VL-7B-Instruct (Bai et al., 2025) as our base model and adpot EasyR1 (Zheng et al., 2025) as our reinforcement learning framework. All experiments are conducted on 8 NVIDIA H20 96G GPUs. We adopt the default settings from EasyR1, using a learning rate of $1e^{-6}$, a global batch size of 128, a rollout batch size of 512, and a rollout size of 8. The analysis of rollout size and different coefficients is provided in Appendix G and Appendix F.

### 4.2 ABLATION STUDIES

To better understand the contribution of each component in our method, we conduct ablation studies on five benchmarks by comparing four settings: vanilla DAPO, DAPO with perceptual regrouping, DAPO with reasoning regrouping, and our final method. Results are summarized in Table 1.

**The effects of Perceptual difficulty-based regrouping.** Using perceptual difficulty-based regrouping alone yields consistent performance gains across benchmarks. For instance, on HallusionBench, which is explicitly designed to evaluate perceptual ability, we observe an improvement of 3.4% over vanilla DAPO. This demonstrates that regrouping samples via spectral analysis of image patch covariances enhances the model's perceptual grounding by mitigating the dominance of extremely easy or hard cases.

**The effects of Reasoning difficulty-based regrouping.** When regrouping with model confidence, the average accuracy increases to 58.4, and it's notable to observe a 3.8% gain on MathVerse, even

Table 2: Performance comparison of Multi-modal LLMs with over 5 benchmarks. Accuracy scores (%) are reported for all benchmarks for clarity. Data sizes used for SFT and RL are annotated in blue and red, respectively. The best value in each column is shown in **bold**, and the second-best is underlined.

| Model | Data Size | MathVerse | MathVision | MathVista | WeMath | HallusionBench | Average |
|---|---|---|---|---|---|---|---|
| *Close-source models* | | | | | | | |
| GPT-4o | - | 50.8 | 30.4 | 63.8 | 69.0 | 71.4 | - |
| Claude-3.5-Sonnet | - | 26.5 | 38.0 | 67.7 | - | 71.6 | |
| *Open-source models* | | | | | | | |
| InternVL-2.5-8B-Instruct (Chen et al., 2024) | - | 39.5 | 19.7 | 64.4 | - | 67.3 | - |
| LLaVA-OneVision-7B (Li et al., 2024) | - | 26.2 | - | 63.2 | - | 48.4 | - |
| Kimi-VL-16B (Du et al., 2025a) | - | 44.9 | 21.4 | 68.7 | - | 66.2 | - |
| URSA-8B (Luo et al., 2025) | - | 45.7 | 26.2 | 59.8 | - | - | - |
| Mulberry-7B (Yao et al., 2024) | - | - | - | 63.1 | - | - | - |
| *reinforcement learning with verifiable reward based* | | | | | | | |
| R1-VL-7B (Zhang et al., 2025a) | 260K+10K | 52.2 | 28.2 | 74.3 | 69.0 | 57.2 | 56.2 |
| Vision-R1-7B (Huang et al., 2025b) | 200K+10K | 52.4 | 27.2 | **73.5** | 62.9 | 69.2 | 57.0 |
| R1-OneVision-7B (Yang et al., 2025b) | 155K+10K | 46.1 | 22.5 | 63.9 | 62.1 | 65.6 | 52.0 |
| OpenVLThinker-7B (Deng et al., 2025) | 35K+15K | 48.0 | 25.0 | 71.5 | 67.8 | 70.8 | 56.6 |
| MM-Eureka-Qwen-7B (Meng et al., 2025) | 15K | 50.5 | 28.3 | 71.5 | 65.5 | 68.3 | 56.8 |
| ADORA-7B (Gui & Ren, 2025) | 2.1K | 50.1 | 27.6 | 71.1 | 67.1 | 53.1 | 53.8 |
| ThinkLite-7B-VL (Wang et al., 2025d) | 11K | 50.2 | 27.6 | 72.7 | 69.2 | 71.0 | 58.1 |
| VLAA-Thinker-7B (Chen et al., 2025a) | 25K | 49.9 | 26.9 | 68.8 | 67.9 | 68.6 | 56.4 |
| NoisyRollout (Liu et al., 2025a) | 2.1K | **53.2** | 28.5 | 72.6 | 69.6 | 72.1 | 59.2 |
| Qwen2.5-VL-7B-Instruct (Bai et al., 2025) | - | 46.2 | 25.0 | 67.5 | 63.1 | 64.6 | 53.3 |
| + Vanilla GRPO | 2.1K (Geometry3K) | 49.6 | 26.8 | 70.2 | 68.2 | 69.8 | 56.9 |
| + Vanilla GRPO + Ours | 2.1K (Geometry3K) | 52.8 | 28.8 | 72.3 | 69.2 | **72.9** | 59.2 |
| + Vanilla DAPO | 2.1K (Geometry3K) | 50.4 | 27.6 | 70.7 | 69.4 | 68.6 | 57.3 |
| + Vanilla DAPO + Ours | 2.1K (Geometry3K) | 51.9 | **29.0** | 72.2 | **71.8** | 71.4 | **59.3** |
| + Vanilla DAPO + Ours | 39K (ViRL39K) | 52.4 | 29.9 | 73.8 | 72.0 | 72.5 | 60.1 |

surpassing the performance of our method. This indicates that the model's internal confidence estimation also serves as a reliable signal for stabilizing optimization.

The combination of both strategies achieves the best overall performance with an average accuracy of 59.3. This confirms that these two strategies provide complementary perspectives on samples, and their integration leads to more robust policy optimization.

### 4.3 COMPARISON WITH STATE-OF-THE-ART APPROACHES

In this subsection, we comprehensively compare our method with various state-of-the-art methods, including closed-source, open-source, and RLVR-related approaches. The experimental results are listed in Table 2. We can draw the following observations: (1) compared with those either distilled from large-scale chain-of-thought data or employing complex data augmentation strategies, our method, utilizing only 2.1k training samples, achieves comparable or even superior performance, significantly demonstrating our effectiveness. (2) Building upon both GRPO and DAPO, our strategy demonstrates promising performance gains. Specifically, we achieve more than 3.5% performance improvements, especially on the HallusionBench; our strategy achieves more than 5% improvements, further showing our effectiveness.

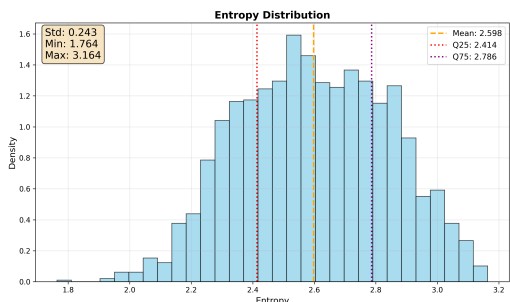

Figure 2: The distribution of pre-calculated entropy on the Geometry3K dataset, where $x$ axis represents entropy, $y$ axis denotes the probability density, Q25 and Q75 denotes 25th and 75th percentiles, respectively .

### 4.4 HYPER-PARAMETER SENSITIVITY ANALYSIS

In this section, we analyze the effects of hyperparameters, including both the number of perceptual and reasoning difficulty-based groups.

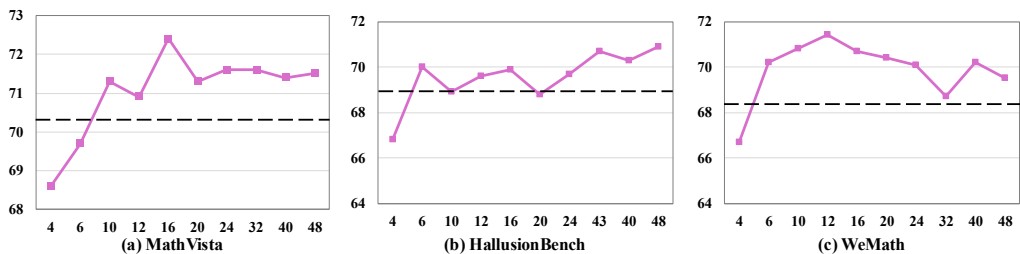

Figure 3: The Accuracy performance of different numbers of groups $b$ on 3 representative benchmarks, where the $x$ axis is the number of groups, and $y$ axis is the results.

### 4.4.1 GROUPS UNDER PERCEPTUAL DIFFICULTY-BASED STRATEGY

As shown in Figure 2 , to regroup samples by entropy, we adopt the 25th and 75th percentiles as thresholds. This quantile-based choice is inherently distribution-aware, as it adapts to the empirical spread of entropy values rather than relying on arbitrary fixed cutoffs. Importantly, it produces a natural 1:2:1 partition of the data—approximately 25% easy, 50% medium, and 25% hard—which avoids the issue of overly sparse or overly dense categories. Such a balance is desirable for stable optimization: each group contains sufficient samples to provide reliable intra-group statistical estimates, while extremely low- and high-entropy cases are isolated rather than allowed to dominate normalization. Moreover, this three-level categorization is semantically interpretable, with low entropy corresponding to simple scenes, high entropy to complex ones, and the middle range capturing moderately difficult cases. Detailed cases representing the entropy of these three categories are illustrated in Appendix H.1.

### 4.4.2 GROUPS UNDER REASONING DIFFICULTY-BASED STRATEGY

We report the effect of varying the number of groups for our reasoning-based strategy in Figure 3. We observe that the performance is relatively stable across a wide range of groups, suggesting that our method is robust to this hyperparameter. For instance, on MathVista and HallusionBench, the accuracy is steadily improved as the number of groups increases to around 12–16, after which the results plateau with only minor fluctuations. A similar trend is observed on WeMath, where the performance peaks at 12 groups but remains competitive even when more groups are introduced.

## 5 RELATED WORKS

In this section, we overview of the related studies. Specifically, we first discuss representative strategy to construct multimodal reasoning models, including chain-of-thought distillation, reinforcement learning, and visual tool integration. Then we introduce the RLVR and its optimization variants, and finally, we highlight the key differences between ours and existing approaches.

### 5.1 MULTIMODAL REASONING MODELS

**Chain-of-thought distillation.** Supervised fine-tuning (SFT) on long chain-of-thought (CoT) data enables models to learn detailed reasoning traces, thereby improving reasoning accuracy. Specifically, building upon (Zhang et al., 2023), this strategy has proven effective through both transferring CoT-enhanced LLMs to multimodal settings (Du et al., 2025b) and training directly with multimodal reasoning data (Liu et al., 2023). Future works futher explore different forms of intermediate reasoning supervision (Dai et al., 2023; Yang et al., 2025c; Zhu et al., 2023).

**Reinforcement learning.** Another line of research leverages reinforcement learning (RL) to optimize reasoning trajectories beyond imitation. Most studies adopt PPO (Schulman et al., 2017) or GRPO (Shao et al., 2024b), with representative approaches such as (Zhang et al., 2025b; Shen et al., 2025; Wang et al., 2025b;c) that apply RL across diverse domains. We will elaborate on RLVR and GRPO in the following subsection(Section 5.2).

**Visual tool integration.** This paradigm moves beyond merely "thinking about images" toward actively querying, modifying, and generating visual information as intermediate steps in reasoning, forming a "visual chain of thought". The development of think-with-image can be roughly divided into three stages (Su et al., 2025) : from external tool exploration (Ma et al., 2024; Shao et al., 2024a; Ma et al., 2025), through programmatic manipulation (Surís et al., 2023; Fu et al., 2025), to intrinsic imagination (Zhao et al., 2025; Chen et al., 2025b). These three stages reflect interconnected capabilities—active exploration, structured reasoning, and generative planning—that together transform visual representations from static inputs into a dynamic workspace for thought.

## 5.2 Reinforcement Learning with Verifiable Reward (RLVR)

RLVR (Lambert et al., 2024) is an optimization paradigm that replaces subjective reward scores with verifiable signals. Its core algorithm, GRPO (Shao et al., 2024b), stabilizes training by comparing candidate responses within a group. Building on GRPO, subsequent studies can be broadly categorized into two directions: data-centric approaches, which expand the candidate and reward space through data manipulation or augmentation, and algorithm-centric approaches, which refine GRPO to strengthen semantic grounding and coherent reasoning.

**Data-centric GRPO.** This line of work enlarges the candidate set (Chen et al., 2025d) or restructures the training data (Chen et al., 2025c; Zhu et al., 2025) so that group comparisons capture richer behaviors. By manipulating data distributions (Zhu et al., 2025) or augmenting inputs (Li et al., 2025; Liu et al., 2025a), these methods expose models to a wider variety of responses, thereby increasing the likelihood of discovering high-quality verifiable signals.

**Algorithm-centric GRPO.** In contrast, algorithm-centric methods refine how verifiable signals guide reasoning. Rather than expanding candidate sets, they adapt GRPO to enhance semantic grounding (Yu et al., 2025a; Liu et al., 2025c) and logical coherence (Huang et al., 2025a; Wei et al., 2025). These approaches emphasize the role of visual grounding and promote reasoning chains where intermediate steps remain verifiable while supporting the final answer.

**Difference.** Compared with existing methods, we regroup the data in advantage calculation based on (1) the model's response uncertainty and (2) the entropy of image inputs when computing the *std*, and sharing the *std* within each group. This design prevents the model from overfitting to extreme samples and enhances its ability to capture the data distinction within each group.

## 6 Conclusion

In this work, we identify a critical challenge in GRPO-based reinforcement learning methods for multimodal reasoning tasks: **the *std*-based group normalization is sensitive to extreme samples**, such as response groups that are almost entirely positive or negative. While this issue exists in GRPO in general, it is significantly amplified in multimodal settings due to the joint influence of perceptual complexity and reasoning uncertainty.

To address this, we propose a simple yet effective **difficulty-aware re-grouping** strategy. By decomposing the sample difficulty into perceptual and reasoning aspects, we construct groups of samples with similar difficulty levels, allowing each group to share *std* during normalization. The normalized advantages with shared *std* from both aspects are combined via element-wise multiplication, effectively integrating data complexity and model uncertainty while preserving intra-group distinctions. By applying over GRPO and DAPO, our strategy achieves more than 2% average performance gains across multiple multimodal reasoning benchmarks.

**Limitations and Future Work.** While difficulty-aware regrouping significantly stabilizes GRPO-based multimodal reasoning, several directions remain open: (1) more precise difficulty estimation —— potentially learned from model behaviors to richer visual-semantic features —— could capture subtler perceptual and reasoning characteristics; (2) adaptive grouping strategies that dynamically adjust group sizes or composition as training progresses may better balance intra-group distinctions and mitigate extreme samples. Beyond technical refinements, the underlying principle of aligning optimization with sample difficulty also offers a general paradigm for stabilizing reinforcement learning optimization with multimodal inputs.

## 7 ETHICS STATEMENT

Our work focuses on algorithmic improvements for reinforcement learning-based multimodal reasoning tasks using publicly available datasets (e.g., MathVista, MathVerse, MathVision). We do not collect data from human subjects, and all datasets used are publicly released. We do not foresee any direct, immediate, or negative societal impacts stemming from the outcomes of our research.

## 8 REPRODUCIBILITY STATEMENT

To ensure reproducibility, we provide detailed descriptions of datasets, training configurations, and hyperparameters in both the main text and supplementary materials. We will provide our code to facilitate reproducibility.

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

## A   THE USE OF LARGE LANGUAGE MODELS(LLMs)

We conducted a study on improving GRPO to further enhance the reasoning capability of MLLMs, achieving substantial performance gains on datasets such as MathVerse (Zhang et al., 2024), MathVision (Wang et al., 2024), and MathVista (Lu et al., 2024). During the preparation of this manuscript, we used LLMs to assist with tasks such as grammar correction, language refinement, and logical checking. However, we confirm that no outputs from the LLMs were directly used; instead, all content underwent careful verification and reconstruction by the authors.

## B   PROMPT DESIGN

We use a "Thinking prompt" to formalize the output of the model. It requires the model to put its reasoning process within <think>...</think> and the final answer in \boxed{}. We keep the system prompt of Qwen2.5-VL (Bai et al., 2025) and prepend the "Thinking prompt" to the user message. The same format is used for both training and evaluation. The full instruction prompt is as follows:

---

**Prompt Example**

**SYSTEM:**
You are a helpful assistant.
**USER:**
You FIRST think about the reasoning process as an internal monologue and then provide the final answer. The reasoning process MUST BE enclosed within <think> </think> tags. The final answer MUST BE put in \boxed{}.<QUESTION>

---

## C  EXPERIMENT SETTINGS

**Reward Calculation.** We adopt a combination of format reward and accuracy reward as the final reinforcement learning signal. The two components are defined as follows:

$$r_{\text{format}} = \begin{cases} 1, & \text{if the output format is correct,} \\ 0, & \text{otherwise,} \end{cases} \tag{21}$$

$$r_{\text{acc}} = \begin{cases} 1, & \text{if the answer matches the ground truth,} \\ 0, & \text{otherwise.} \end{cases} \tag{22}$$

The overall reward is computed as the weighted sum of the two:

$$r_{\text{overall}} = 0.1 \times r_{\text{format}} + 0.9 \times r_{\text{acc}}. \tag{23}$$

A smaller weight is assigned to the format reward, since response formatting is relatively easy to learn compared with accuracy.

## D  ANALYSIS OF THE OCCURRENCE OF EXTREME SAMPLES, WHICH *std*-BASED GROUP NORMALIZATION IS HIGHLY SENSITIVE TO.

To further support the motivation of our paper and analyze the changes of extreme samples during training, we perform a detailed step-by-step analysis of reward statistics across 60 training steps with 512 samples and their 8 rollout rewards. The empirical evidence clearly shows that the existence of extreme samples is not an occasional event but a persistent and systemic phenomenon.

Table 3: The statistics of rewards about extreme samples within the batch across 60 training steps.

| Training steps | 1 | 10 | 20 | 30 | 40 | 50 | 60 |
| --- | --- | --- | --- | --- | --- | --- | --- |
| Effective samples (participating in training) | 323 | 327 | 324 | 322 | 297 | 314 | 306 |
| Extreme success (7 correct & 1 wrong) | 41 | 39 | 48 | 66 | 78 | 60 | 82 |
| Extreme failure (7 wrong & 1 correct) | 78 | 89 | 74 | 51 | 54 | 54 | 51 |
| Total Extreme Ratio | 36.8% | 39.1% | 37.7% | 36.3% | 44.4% | 36.31% | 43.5% |

First, groups with 8 identical rewards (i.e., variance = 0) constitute 35%–46% of all samples at every training step. We first exclude these groups for not participating in gradient updates.

Second, during the training process, there are **31%–44% samples exhibit the 7:1 extreme reward patterns** (i.e., 7/8 correct or wrong) among the remaining effective samples, which produce extremely small variance. Besides, the occurrence of this situation will increase as training deepens.

These findings demonstrate that **the instability of *std*-based normalization is structural rather than incidental:** multimodal reasoning tasks naturally contain a large proportion of very easy and very hard samples, leading to unstable and unreliable advantage scaling. This directly motivates our difficulty-aware regrouping strategy, which stabilizes normalization by ensuring that variance is computed only within samples of comparable difficulty.

## E  VERIFY THE FEASIBILITY OF UTILIZING IMAGE ENTROPY AS A PROXY FOR PERCEPTUAL DIFFICULTY AND MODEL CONFIDENCE AS A PROXY FOR REASONING DIFFICULTY.

### E.1  IMAGE ENTROPY AS A PROXY FOR PERCEPTUAL DIFFICULTY.

**Perceptual difficulty** in our framework is defined based on **the complexity of visual embeddings**, which we quantify using spectral analysis of image patch covariances. Specifically, the entropy of the eigenvalue distribution from the covariance matrix reflects the amount of variance across spatial features in the image. Researchers in prior works(Grzywacz, 2025) support that: high entropy

indicates a more diverse distribution of visual features, implying a richer and more complex visual structure. This complexity makes it more challenging for the visual model to recognize, and thus, we associate higher entropy with greater perceptual difficulty.

### E.2 MODEL CONFIDENCE AS A PROXY FOR REASONING DIFFICULTY.

Researchers in (Farquhar et al., 2024; Nguyen et al., 2025) propose that "one measure of uncertainty is the predictive entropy of the output distribution, which measures the information one has about the output given the input[3]. The predictive entropy for an input sentence $\mathbf{x}$ is the conditional entropy ($H$) of the output random variable ($Y$) with realization $y$ given $\mathbf{x}$."

$$\mathrm{PE}(\mathbf{x}) = H(Y|\mathbf{x}) = -\sum_{y} P(y|\mathbf{x})\ln P(y|\mathbf{x}). \tag{24}$$

Researchers(Kadavath et al., 2022) also hypothesize that when a model knows the answer to a particular question, it is confident in its response, and this would result in an answer distribution with small entropy. Conversely, when a model is unsure about its response, it will lead to an answer distribution with high entropy, thus implying a more challenging reasoning process.

This aligns directly with our formulation: the **sequence-level log probabilities** we compute are theoretically linked to the notion of *semantic entropy* and represent the joint likelihood of the entire reasoning chain. A low log-probability corresponds to a flat or high-entropy output distribution, reflecting uncertainty in the reasoning trajectory, while a high log-probability corresponds to a confident, low-entropy distribution.

### E.3 EMPIRICAL VALIDATION.

During the evaluation stage, we conduct an analysis focusing on the **questions that the model answered incorrectly** on two benchmarks. We want to examine whether these error samples are concentrated in the more difficult groups as defined by our difficulty metrics. The intuition behind this approach is that samples belonging to higher-difficulty groups—whether in terms of perceptual complexity or reasoning uncertainty—should naturally be harder for the model to tackle. Consequently, we expect these samples to exhibit higher error rates.

To achieve this, we use Gemini2.5 Pro to classify the sources of errors, distinguishing between **perceptual errors** and **reasoning errors**.

- For **perceptual errors**, we first group the images based on their visual entropy, then compute the proportion of incorrect answers within each group relative to the total number of perceptual errors.
- Similarly, for **reasoning errors**, we group the samples based on model confidence, and calculate the proportion of incorrect answers in each group relative to the total number of reasoning errors.

Table 4: The error rate of perceptual difficulty groups in perceptual errors on two benchmarks.

|  | low-entropy | medium-entropy | high-entropy |
|---|---|---|---|
| Wemath | 23.6% | 31.3% | 45.1% |
| HallusionBench | 21.2% | 29.6% | 49.2% |

Table 5: The error rate of reasoning difficulty groups in reasoning errors on two benchmarks.

|  | group 1 (low confidence) | group 2 | group 3 | group 4 | group 5 | group 6 | group 7 | group 8 | group 9 | group 10 (high confidence) |
|---|---|---|---|---|---|---|---|---|---|---|
| Wemath | 13.4% | 12.6% | 13.2% | 12.7% | 11.6% | 9.1% | 9.7% | 7.2% | 6.7% | 4.2% |
| HallusionBench | 12.7% | 11.9% | 11.2% | 11.7% | 10.0% | 9.8% | 9.0% | 8.2% | 8.3% | 7.1% |

As shown in Table 4 and Table 5, the results align with our expectations: images with **low visual entropy** (indicating simplicity) correspond to **lower perceptual error rates**, and samples with **lower**

**model confidence** (indicating greater uncertainty in the reasoning process) correspond to **higher reasoning error rates**. Our empirical findings are consistent with this intuition, further supporting the validity of our difficulty metrics.

The instructions for Gemini2.5 pro to conduct classification are as follows:

---

**Prompt Example**

**SYSTEM:**
You are an expert evaluator for multimodal reasoning errors. Your task is to determine why a model answered a visual reasoning question incorrectly.
You must classify the error into one and only one of the following categories:

1. Perception Error
The model misunderstood or misread the visual content.

Examples include:
Misidentifying objects, numbers, angles, or relations in the image
Failing to notice geometric constraints
Incorrectly interpreting positions, shapes, or labels
Making a wrong assumption about what is visually shown

2. Reasoning Error The model correctly interpreted the visual content, but its logical reasoning or mathematical deduction is incorrect.

Examples include:
Using the wrong theorem or property
Incorrect algebraic or geometric steps
Incorrect reasoning chain
Logical inconsistency

You must output your judgment in JSON format:
{
"error_type":"Perception" or "Reasoning",
"explanation": "A short explanation of why this classification fits."
}

**USER:**
You are given a visual reasoning question, the model's prediction (including its reasoning process), and the correct answer.
Please carefully read the model's reasoning and determine whether the model failed due to Perception Error or Reasoning Error.
Here is the data sample:
<insert the JSON sample here>
Now analyze the model's reasoning step-by-step and output the final JSON in the required format.

---

## F    ANALYSIS OF DIFFERENT WEIGHTING COEFFICIENTS.

In this section, we will analyze the effects of the weighting coefficients for further assessing hyperparameter sensitivity. We experiment with different combinations of three coefficients $\alpha_{Ori}$, $\alpha_{Percep}$, and $\alpha_{Reason}$, and the results are presented in Table 6.

We can observe that while the performance on different benchmarks varies slightly with different settings, the method is relatively stable across a wide range of settings, with no significant degradation in results, indicating that our model is not overly sensitive to the specific choice of hyperparameters. This suggests that our method does not require extremely fine-tuned hyperparameters to perform effectively. Besides, we find that, usually, a little $\alpha_{Ori}$ and relatively larger $\alpha_{Percep}$ and $\alpha_{Reason}$ are more likely to achieve a better effect, which can be used as guidance in the application of our method.

Table 6: The effects of different weighting coefficients (built upon DAPO) on 5 benchmarks

| $\alpha_{Ori}$ | $\alpha_{Percep}$ | $\alpha_{Reason}$ | MathVerse | MathVision | MathVista | WeMath | HallusionBench |
|---|---|---|---|---|---|---|---|
| 0.1 | 0.2 | 0.7 | 50.7 | 28.6 | 71.6 | 71.8 | 71.2 |
| 0.15 | 0.25 | 0.6 | 50.8 | 29.0 | 71.6 | 70.6 | 70.8 |
| 0.2 | 0.1 | 0.7 | 51.2 | 28.4 | 71.5 | 70.4 | 71.0 |
| 0.3 | 0.1 | 0.6 | 51.7 | 27.8 | 70.7 | 70.6 | 70.8 |
| 0.4 | 0.3 | 0.3 | 51.9 | 27.9 | 71.4 | 70.5 | 71.1 |
| 0.6 | 0.2 | 0.2 | 50.4 | 28.8 | 72.2 | 71.0 | 71.4 |
| 0.7 | 0.1 | 0.2 | 51.1 | 28.3 | 70.3 | 71.1 | 69.8 |

## G   ANALYSIS OF THE EFFECT OF ROLLOUT SIZE ON PERFORMANCE AND STABILITY.

Table 7: The effects of rollout size (built upon DAPO) on 5 benchmarks.

| rollout | MathVerse | **MathVision** | MathVista | WeMath | HallusionBench |
|---|---|---|---|---|---|
| 2 | 48.7 | 27.1 | 70.1 | 69.7 | 67.3 |
| 4 | 50.1 | 28.4 | 71.5 | 70.2 | 68.9 |
| 8 | 51.9 | 29.0 | 72.2 | 71.8 | 71.4 |
| 16 | 52.1 | 29.2 | 72.1 | 71.2 | 71.2 |
| 24 | 51.7 | 29.0 | 71.9 | 71.9 | 71.5 |
| 32 | 51.9 | 28.9 | 72.2 | 71.0 | 71.3 |

We observed from Table 7 that when the rollout size is smaller than 8, the performance improves as the rollout size increases. Notably, when the number of rollouts is reduced to 2, the model reverts to PPO. As the rollout size continues to increase beyond 8, the improvement in performance becomes less pronounced, eventually stabilizing at a stable value.

These results indicate that while increasing the number of rollouts can lead to better performance, after a certain point, beyond which further increases in group size do not significantly contribute to performance improvement. This shows the importance of selecting an appropriate group size to balance computational cost and model performance.

# H CASE STUDY

## H.1 PERCEPTUAL DIFFICULTY-BASED RE-GROUPING CASES

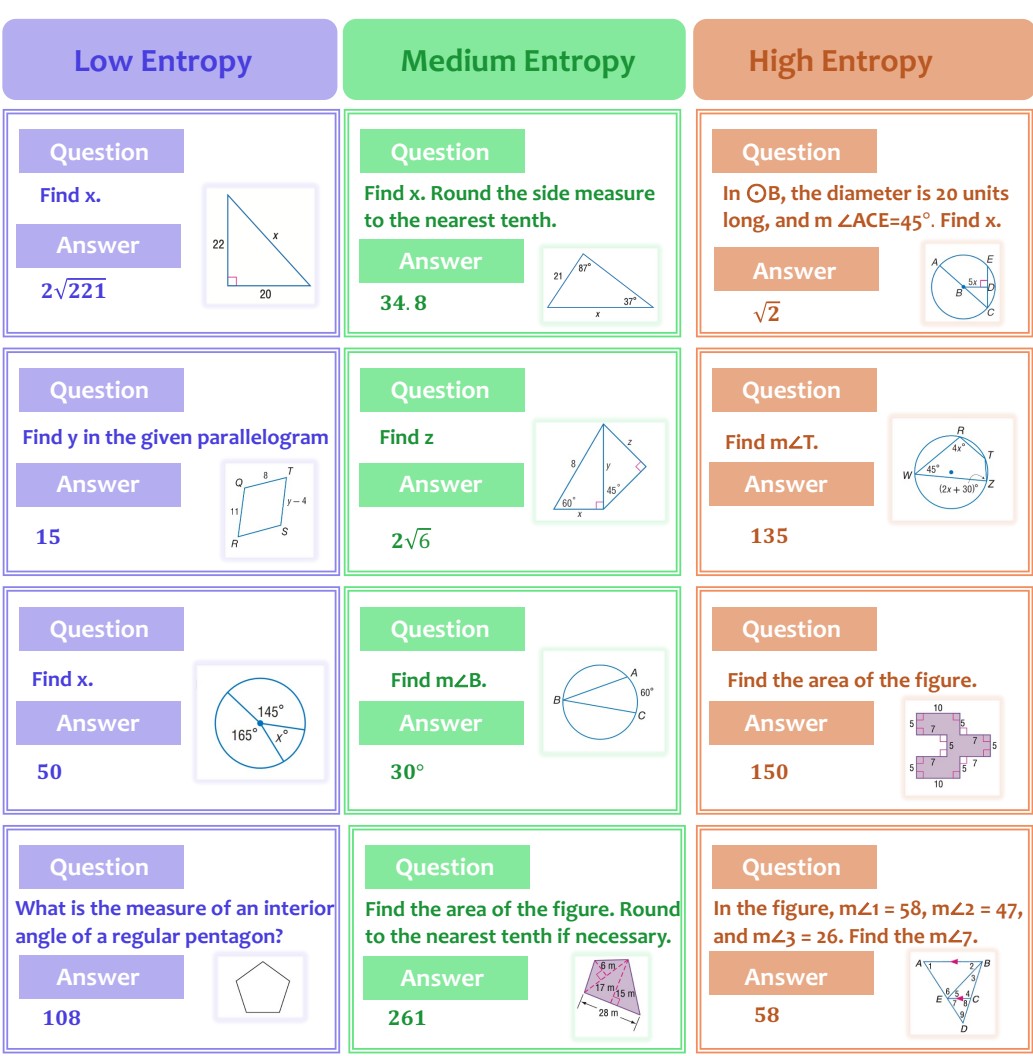

Figure 4: Illustrative examples of different levels of entropy

## H.2 DEMONSTRATION OF IMPROVED PERCEPTION AND REASONING CAPABILITIES

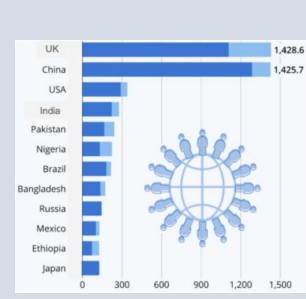

**Question:**
According to the chart, does China have the second largest population in the world? You final answer can only be \boxed{yes} or \boxed{no}.

**Source:** HallusionBench
**Problem version:** chart
**figure id:** 3_2
**question_id:** 3

### Ours:

<think>
The chart represents population data for various countries, with China and the UK having the highest populations at 1,425.7 and 1,428.6 million respectively.
Since the chart indicates China's population is slightly lower than the UK's, it suggests that China does indeed have the second largest population in the world according to this data.
</think>
\boxed{yes}

### Vanilla GRPO:

<think>
The chart shows the population of various countries. The population of China is listed as 1,425.7 million, and the population of the UK is listed as 1,428.6 million. Since the UK has a slightly higher population, China does not have the second largest population in the world. The second largest population in the world is the UK.
</think>
\boxed{no}

Figure 5: Case Study1 showing improved reasoning capability over vanilla GRPO

1080
1081
1082
1083
1084
1085
1086
1087
1088
1089
1090
1091
1092
1093
1094
1095
1096
1097
1098
1099
1100
1101
1102
1103
1104
1105
1106
1107
1108
1109
1110
1111
1112
1113
1114
1115
1116
1117
1118
1119
1120
1121
1122
1123
1124
1125
1126
1127
1128
1129
1130
1131
1132
1133

**Question:**
Each of the cubes in the figure has the length of an edge equal to 1. What is the length of the segment AB?
(A) $\sqrt{17}$ (B) 7 (C) $\sqrt{13}$ (D) $\sqrt{7}$ (E) $\sqrt{14}$

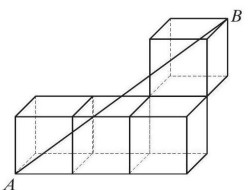

**Source:** Mathvision
**question_id:** 209

## Ours:

\<think\>
To find the length of the segment AB, we can use the three-dimensional distance formula, which is an extension of the Pythagorean theorem. The distance d between two points $(x_1, y_1, z_1)$ and $(x_2, y_2, z_2)$ in three-dimensional space is given by: $d = \sqrt{(x_1 - x_2)^2 + (y_1 - y_2)^2 + (z_1 - z_2)^2}$
Here, point A is at (0, 0, 0) and point B is at (3, 2, 2). Plugging these coordinates into the formula, we get:
$AB = \sqrt{(3 - 0)^2 + (2 - 0)^2 + (2 - 0)^2} = \sqrt{3^2 + 2^2 + 2^2} = \sqrt{17}$
Therefore, the length of the segment AB is $\sqrt{17}$.
\</think\>
\boxed{A}

## Vanilla GRPO:

\<think\>
To find the length of the segment AB, we need to consider the three-dimensional space formed by the cubes. The segment AB can be visualized as a diagonal that spans across three unit cubes in the x, y, and z directions.
1. The segment AB moves 3 units in the x-direction, 2 units in the y-direction, and 1 unit in the z-direction.
2. The length of AB can be found using the three-dimensional distance formula, which is similar to the Pythagorean theorem but extended to three dimensions: $d = \sqrt{x^2 + y^2 + z^2}$
3. Substituting the values x = 3, y = 2, and z = 1: $AB = \sqrt{3^2 + 2^2 + 1^2} = \sqrt{14}$
Therefore, the length of the segment AB is $\sqrt{14}$
\</think\>
\boxed{$\sqrt{14}$}

Figure 6: Case Study2 showing improved reasoning capability over vanilla GRPO

**Question:**
As shown in the figure, AB is the diameter of
⊙ O, CD is the chord of ⊙ O, ∠ADC = 26.0, then
the degree of ∠CAB is ():
(A) 26° (B) 74° (C) 64° (D) 54°

**Source:** Mathvista
**question_id:** 59

### Ours:

<think>
Since AB is the diameter of circle O, angle ACB is a right angle (90°) because any angle inscribed in a semicircle is a right angle. We are given that angle ADC is 26°. Because angles subtended by the same arc are equal, angle ACB (which is 90°) and angle ADB (which is also 26°) are related to the arc AB. However, to find angle CAB, we can use the fact that the sum of angles in triangle ACB is 180°. Since angle ACB is 90° and angle ABC (opposite to side AC) is the same as angle ADC (26°) due to the inscribed angle theorem, we can calculate angle CAB as follows:
angle CAB = 180° - angle ACB - angle ABC
angle CAB = 180° - 90° - 26°
angle CAB = 64°
Therefore, the degree of angle CAB is 64°.
</think>
\boxed{C}

### Vanilla GRPO:

<think>
Since AB is the diameter of circle O, angle ACB is a right angle (90 degrees) because the angle subtended by a diameter in a semicircle is always 90 degrees. Given that angle ADC is 26 degrees, and angles ADC and CAB are subtended by the same arc AC, they are equal. Therefore, angle CAB is also 26 degrees.
So, the degree of angle CAB is 26 degrees.
</think>
\boxed{A}

Figure 7: Case Study3 showing improved reasoning capability over vanilla GRPO

**Question:**
As shown in the diagram, in a building, it is known that A lives to the left of B. Who lives below D?
A. A;   B. E;   C. F;   D. G;   E. No correct answer

| A | B | C |
|---|---|---|
| D | E | F |
| G | H | I |

**Source:** Wemath
**question_id:** 2steps_17@2steps_1

**Ours:**

<think>
The diagram shows a grid with letters A through I. The question states that A lives to the left of B. This means that if A is in a certain position, B must be to the right of A in the same row. However, the question asks who lives below D. Since D is in the second row, the person below D would be in the third row, which is G.
So, the person who lives below D is G.
</think>
\boxed{D}

**Vanilla GRPO:**

<think>
The diagram shows a grid with letters A through I. The question states that A lives to the left of B. This means that if we look at the horizontal row where A is located, B must be to the right of A. Now, we need to determine who lives below D.
In the grid:
- D is in the second row, first column.
- The letter below D is E.
Therefore, the person who lives below D is E.
</think>
\boxed{B}

Figure 8: Case Study4 showing improved reasoning capability over vanilla GRPO

