# OpenReview forum: "Enhancing Multi-Modal LLMs Reasoning via Difficulty-Aware Group Normalization"
_ICLR.cc/2026/Conference — Submitted to ICLR 2026_

### Official Review · Reviewer_Qc3t · 2025-10-29

**Soundness:** 3
**Presentation:** 3
**Contribution:** 3
**Rating:** 8
**Confidence:** 3

**Summary:**

This paper addresses the instability of standard deviation-based normalization in GRPO for multimodal LLMs, where extreme samples with nearly uniform rewards distort training. The authors propose difficulty-aware group normalization that re-groups samples based on visual entropy and model confidence. The method preserves intra-group distinctions while eliminating sensitivity to extreme cases.

**Strengths:**

1. The paper identifies a clear limitation in existing GRPO methods: std-based normalization becomes unstable when groups contain mostly positive or negative rewards.
2. Based on the identified limitation, the paper designs a interest algorithm that leverages the characteristics of multimodal models by performing additional grouping based on model uncertainty and inherent image complexity to calculate advantages. This algorithm demonstrates novelty in addressing multimodal-specific challenges.
3. The algorithm's approach is not limited to multimodal settings, and instead of simply expanding rollout sizes, the paper proposes an method for forming groups based on additional criteria. This may be interested to community for further exploration in various domains.

**Weaknesses:**

1. The method introduces more hyperparameters that may require careful tuning, including the weighting coefficients (α_Ori, α_Percep, α_Reason) and the percentile thresholds for grouping.

**Questions:**

1. How do the authors validate that their difficulty metrics actually capture the intended concepts?
2. Could the authors provide experimental results about hyperparameter sensitivity?

---

> ### Author Response · Authors · 2025-11-26
> **Response to Reviewer Qc3t ---- Part 1/2**
>
> **Response to Reviewer $\color{blue}{\text{Qc3t}}$**
>
> We highly appreciate your invaluable comments and positive feedback of our work, which inspires us to greatly improve our paper! To address your concerns, we present the point-to-point responses as follows. We will carefully revise our paper, taking all your feedbacks into account.
>
> > **Comment 1 + Question2: Experimental results about hyperparameter sensitivity** - "The method introduces more hyperparameters that may require careful tuning ... ?", "Could the authors provide experimental results about hyperparameter sensitivity?"
>
> Thank you for raising this point! We fully understand your concerns and we think it's necessary to discuss them in the paper. To assess hyperparameter sensitivity, we analyze the effects of the weighting coefficients ($\alpha_{Ori}, \alpha_{Percep}, \alpha_{Reason}$), and the percentile thresholds for grouping:
>
> - **The weighting coefficients**
>
> We experimented with different combinations of three coefficients, and the results are presented in Table 1.
>
> **Table 1: The effects of different weighting coefficients of our method (build upon DAPO) on 5 benchmarks.**
>
> | $\alpha_{Ori}$ | $\alpha_{Percep}$ | $\alpha_{Reason}$ | MathVerse | **MathVision** | MathVista | WeMath | HallusionBench |
> | :------------: | :---------------: | :---------------: | :-------: | :------------: | :-------: | :----: | :------------: |
> |      0.1       |        0.2        |        0.7        |   50.7    |      28.6      |   71.6    |  71.8  |      71.2      |
> |      0.15      |       0.25        |        0.6        |   50.8    |      29.0      |   71.6    |  70.6  |      70.8      |
> |      0.2       |        0.1        |        0.7        |   51.2    |      28.4      |   71.5    |  70.4  |      71.0      |
> |      0.3       |        0.1        |        0.6        |   51.7    |      27.8      |   70.7    |  70.6  |      70.8      |
> |      0.4       |        0.3        |        0.3        |   51.9    |      27.9      |   71.4    |  70.5  |      71.1      |
> |      0.6       |        0.2        |        0.2        |   50.4    |      28.8      |   72.2    |  71.0  |      71.4      |
> |      0.7       |        0.1        |        0.2        |   51.1    |      28.3      |   70.3    |  71.1  |      69.8      |
>
> We can observe that while the performance on different benchmarks varies slightly with different settings, the method is relatively stable across a wide range of settings, with no significant degradation in results, indicating that our model is not overly sensitive to the specific choice of hyperparameters. This suggests that our method does not require extremely fine-tuned hyperparameters to perform effectively. Besides, we find that, usually a little $\alpha_{Ori}$ and relatively lager $\alpha_{Percep}$ and $\alpha_{Reason}$ are more likely to achieve a better effect, which can be used as a guidance in the application of our method.
>
> - **The percentile thresholds for grouping**
>   - Perceptual difficulty-based regrouping: as shown in Figure 2 of our paper, to regroup samples by entropy, we adopt the 25th and 75th percentiles as thresholds. This quantile-based approach is distribution-aware, as it adapts to the empirical spread of entropy values rather than relying on arbitrary fixed cutoffs. The use of percentiles as natural thresholds reflects the inherent distribution of perceptual difficulty in the data. Importantly, the 25th and 75th percentiles produce a natural 1:2:1 partition of the data—approximately 25% easy, 50% medium, and 25% hard—which ensures a balanced grouping, avoiding the problem of overly sparse or overly dense categories. We also represent the Perceptual difficulty-based group cases in `Appendix D.1`, where we can observe that images with different levels of entropy demonstrate different levels of visual complexity.
>   - Reasoning difficulty-based regrouping: we conduct an extensive sensitivity analysis by varying the number of groups $b\in\{4,6,8,10,12,16,20,24,32,40,48\}$, and Figure 3 of our paper summarizes the results on MathVista, HallusionBench, and WeMath. We can find that:
>     - Performance is highly stable across a wide range of group counts. For 10 to 48 groups, all benchmarks fluctuate within a narrow band, demonstrating low sensitivity to this hyperparameter.
>     - A consistent performance peak emerges around 12–16 groups. This range corresponds to a balance avoiding both under and over partitioning.
>
> Thank you again for carefully pointing them out! We will enrich the analysis in `Section 4.4.1 & 4.4.2`, and supplement the discussion of different weighting coefficients in our revised version.

---

> > ### Comment · Reviewer_Qc3t · 2025-11-26
> >
> > Thank you for the response and additional experiments.
> >
> > While your hyperparameter analysis is thorough, I notice you haven't tested the extreme cases where one coefficient equals 1 and others equal 0 (e.g., α_Ori = 1, α_Percep = 0, α_Reason = 0). Although these may not reflect your original intended design, such ablation studies would help understand the influence of each grouping strategy.
> >
> > Introducing additional grouping presents novelty to me personally, and I do not see other reviewers identify similar work. Given my positive overall assessment, I will increase the contribution score while maintaining the overall rating.

---

> > > ### Author Response · Authors · 2025-11-28
> > > **Response to new questions and thanks reviewer.**
> > >
> > > Dear Reviewer $\color{blue}{\text{Qc3t}}$,
> > >
> > > We greatly appreciate your recognition of our efforts and increasing the contribution score. Your support is our greatest motivation!
> > >
> > > Thanks for pointing out this situation! We fully agree that testing three extreme cases can further clarify the contribution of each grouping strategy. We find that: When
> > >
> > > (1) $\alpha_{Ori}=1, \alpha_{Percep}=\alpha_{Reason}=0$, only DAPO was used.
> > >
> > > (2) $\alpha_{Percep}=1, \alpha_{Ori}=\alpha_{Reason}=0$ means only the strategy of perceptual regrouping is employed upon DAPO.
> > >
> > > (3) $\alpha_{Reason}=1, \alpha_{Ori}=\alpha_{Percep}=0$ means only the strategy of reasoning regrouping is employed upon DAPO.
> > >
> > > These three cases are consistent with the ablation study in Table 1 of our paper, and the results are as follows:
> > >
> > > **Table 4: The effects of the extreme coefficient of our method (build upon DAPO) on 5 benchmarks.**
> > >
> > > | $\alpha_{Ori}$ | $\alpha_{Percep}$ | $\alpha_{Reason}$ | MathVerse | **MathVision** | MathVista | WeMath | HallusionBench |
> > > | :------------: | :---------------: | :---------------: | :-------: | :------------: | :-------: | :----: | :------------: |
> > > |       1        |         0         |         0         |   50.4    |      27.6      |   70.7    |  69.4  |      68.6      |
> > > |       0        |         1         |         0         |   52.3    |      28.0      |   71.4    |  70.9  |      70.9      |
> > > |       0        |         0         |         1         |   52.3    |      28.3      |   71.6    |  70.8  |      68.9      |
> > >
> > > The findings show that each component contributes complementary benefits, and none of the single-coefficient settings match the performance of the full formulation. This supports the necessity of combining the original advantage with both perceptual and reasoning difficulty normalization.
> > >
> > > Thank you once again for your valuable time and effort in reviewing our paper!
> > >
> > > Best regards,
> > >
> > > Authors of 17401

---

> ### Author Response · Authors · 2025-11-26
> **Response to Reviewer Qc3t ---- Part 2/2**
>
> > **Question 1: Validate the difficulty metrics** - "How do the authors validate that their difficulty metrics actually capture the intended concepts?"
>
> Thank you for drawing our attention to this important aspect! In light of this, we further conduct some statistical experiments for validation.
>
> During the evaluation stage, we conduct an analysis focusing on the **questions that the model answered incorrectly** on two benchmarks. We want to examine whether these error samples are concentrated in the more difficult groups as defined by our difficulty metrics. The intuition behind this approach is that samples belonging to higher-difficulty groups—whether in terms of perceptual complexity or reasoning uncertainty—should naturally be harder for the model to tackle. Consequently, we expect these samples to exhibit higher error rates.
>
> To achieve this, we use Gemini2.5 Pro to classify the sources of errors, distinguishing between **perceptual errors** and **reasoning errors**. The instructions for Gemini2.5 pro to conduct classification are as follows:
>
> ```
> ########################## Prompt Example ##########################
> System Prompt：
>
> You are an expert evaluator for multimodal reasoning errors. Your task is to determine why a model answered a visual reasoning question incorrectly.
> You must classify the error into one and only one of the following categories:
>
> 1. Perception Error
> The model misunderstood or misread the visual content.
>
> Examples include:
> Misidentifying objects, numbers, angles, or relations in the image
> Failing to notice geometric constraints
> Incorrectly interpreting positions, shapes, or labels
> Making a wrong assumption about what is visually shown
>
> 2. Reasoning Error
> The model correctly interpreted the visual content, but its logical reasoning or mathematical deduction is incorrect.
>
> Examples include:
> Using the wrong theorem or property
> Incorrect algebraic or geometric steps
> Incorrect reasoning chain
> Logical inconsistency
>
> You must output your judgment in JSON format:
> {
>   "error_type": "Perception" or "Reasoning",
>   "explanation": "A short explanation of why this classification fits."
> }
>
>
> User Prompt：
>
> You are given a visual reasoning question, the model’s prediction (including its reasoning process), and the correct answer.
> Please carefully read the model's reasoning and determine whether the model failed due to Perception Error or Reasoning Error.
> Here is the data sample:
> <insert the JSON sample here>
> Now analyze the model's reasoning step-by-step and output the final JSON in the required format.
>
> ```
>
> - For **perceptual errors**, we first group the images based on their visual entropy, then compute the proportion of incorrect answers within each group relative to the total number of perceptual errors.
> - Similarly, for **reasoning errors**, we group the samples based on model confidence, and calculate the proportion of incorrect answers in each group relative to the total number of reasoning errors.
>
> **Table 2: The error rate of perceptual difficulty groups in perceptual errors on two benchmarks.**
>
> |                | low-entropy | medium-entropy | high-entropy |
> | :------------: | :---------: | :------------: | :----------: |
> |     Wemath     |    23.6%    |     31.3%      |    45.1%     |
> | HallusionBench |    21.2%    |     29.6%      |    49.2%     |
>
> **Table 3: The error rate of reasoning difficulty groups in reasoning errors on two benchmarks**.
>
> |                | group 1 (low confidence) | group 2 | group 3 | group 4 | group 5 | group 6 | group 7 | group 8 | group 9 | group 10 (high confidence) |
> | :------------: | :----------------------: | :-----: | :-----: | :-----: | :-----: | :-----: | :-----: | :-----: | :-----: | :------------------------: |
> |     Wemath     |          13.4%           |  12.6%  |  13.2%  |  12.7%  |  11.6%  |  9.1%   |  9.7%   |  7.2%   |  6.7%   |            4.2%            |
> | HallusionBench |          12.7%           |  11.9%  |  11.2%  |  11.7%  |  10.0%  |  9.8%   |  9.0%   |  8.2%   |  8.3%   |            7.1%            |
>
> As shown in **Tables 2 and 3**, the results align with our expectations: images with **low visual entropy** (indicating simplicity) correspond to **lower perceptual error rates**, and samples with **lower model confidence** (indicating greater uncertainty in the reasoning process) correspond to **higher reasoning error rates**. Our empirical findings are consistent with this intuition, further supporting the validity of our difficulty metrics.

---

### Official Review · Reviewer_jCT4 · 2025-10-30

**Soundness:** 3
**Presentation:** 3
**Contribution:** 2
**Rating:** 4
**Confidence:** 4

**Summary:**

This paper observes that in the MLLM RL training process, the limited number of rollouts causes the sampling distribution of the advantage to be significantly affected by variance fluctuations. To address this, the paper proposes grouping images and model-generated texts according to different levels of “difficulty,” and replacing group-level variance estimation with batch-level variance computation.

**Strengths:**

- This paper identifies the abnormal impact of variance on advantage estimation in MLLM reinforcement learning when the number of rollouts is insufficient, and proposes several mitigation strategies.
- It achieves accuracy improvements on MathVista, MathVision, and WeMath benchmarks.
- It attains comparable or superior performance to prior work while using only 2K training samples.

**Weaknesses:**

- The core idea of the paper is to address the bias in reward normalization under a limited number of rollouts by combining batch-level processing with difficulty grouping. However, it does not compare with methods that directly modify the normalization strategy, such as batch-level normalization (e.g., Reinforce++) or no normalization (e.g., Dr.GRPO).
- It remains unclear how performance differs when the number of rollouts increases or decreases (e.g., from 8 to 32 or 2), where the variance effect would be reduced. If the main motivation lies in normalization bias, a comparison with simply increasing the number of rollouts is necessary.
- The baseline results reported in Table 2 are problematic — for instance, R1-VL-7B is based on Qwen2, not Qwen2.5, which may cause confusion, and although Vision-R1-7B provides an available checkpoint, its evaluation results are not reported by the authors.
- The values of the three α-weights in Equation (20) are not specified in either the main text or the appendix, and the paper lacks ablation studies or comparisons analyzing different weight settings.

**Questions:**

- The normalization bias may evolve during training — for example, as training progresses, cases with “8 rollouts but only 1 incorrect answer” become increasingly common. It remains unclear how the Perceptual and Reasoning difficulty groups change throughout the training process.
- Compared with group-level variance computation, the paper does not provide sufficient analysis of how the training dynamics (e.g., reward and advantage trajectories) behave. How does the advantage fluctuation compare to the original approach?
- Table 2 shows that the proposed method achieves greater improvements with fewer data samples. Would performance continue to increase if the dataset size were scaled up to over 10K samples? If not, does this imply that the selected Geometry subset is particularly favorable for achieving high scores on these benchmarks?
- Figure 3 presents results across 4–48 reasoning difficulty groups, but given the batch size, when the number of groups exceeds 40, each group may contain only a few samples. Could this introduce statistical bias? If so, might the observed improvement on HallusionBench be attributed to enhanced self-confidence effects rather than genuine reasoning gains?

---

> ### Author Response · Authors · 2025-11-26
> **Response to Reviewer jCT4 ---- Part 1/4**
>
> **Response to Reviewer $\color{orange}{\text{jCT4}}$**
>
> We highly appreciate your insightful comments, which help us a lot to better scrutiny and polish our work! To address your concerns, we have detailed our responses point-to-point below.
>
> > **Comment 1 + Comment 3: Experimental comparison** - "compare with methods that directly modify the normalization strategy ...",  "The baselines results reported in Table 2 ..."
>
> We first reproduce Reinforce++ and the idea of Dr.GRPO in the framework of EasyR1[1]. As a variant of PPO, Reinforce++ utilizes batch-level normalization when computing advantage, while Dr.GRPO eliminates the standard deviation from GRPO’s advantage computation, addressing instability caused by near-zero std in extreme groups. The experimental results are as follows.
>
> **Table 1: Performance comparisons (build upon DAPO) over 5 benchmarks.**
>
> |              | MathVerse | MathVision | MathVista | wemath | HallusionBench |
> | :----------: | :-------: | :--------: | :-------: | :----: | :------------: |
> |     DAPO     |   50.4    |    27.6    |   70.7    |  69.4  |      68.6      |
> | Reinforce++  |   43.3    |    27.5    |   68.8    |  64.9  |      69.0      |
> |   Dr.GRPO    |   50.9    |    28.1    |   70.3    |  69.1  |      70.2      |
> |   R1-VL-7B   |   52.2    |    28.2    |   74.3    |  69.0  |      57.2      |
> | Vision-R1-7B |   52.4    |    27.2    |   73.5    |  62.9  |      69.2      |
> |     Ours     |   51.9    |    29.0    |   72.2    |  71.8  |      71.4      |
>
> We can observe that:
>
> - For Reinforce++, batch-level normalization provides a stable and theoretically sound, effectively unbiased estimate. However, in multimodal reasoning, images in the same batch may have drastically different perceptual and reasoning difficulty. Batch-level normalization mixes easy and hard samples, leading to distorted advantage scales.
>
> - Dr.GRPO can avoid exploding advantages due to low-std groups, however, it simultaneously discards the valuable intra-group distinctions, which are essential for effective and stable optimization. Therefore, Dr.GRPO indeed achieves slightly better performance than vanilla DAPO on several benchmarks, but still underperforms our method across all five benchmarks.
>
> Besides, thanks for your kind suggestions, and sorry for such confusion! We just followed the experimental results of NoisyRollout[2] in NeurIPS 2025, SynthRL[3] in ICML 2025 in the previous version. We have followed your suggestions to add the results of Vision-R1-7B in the above table 1, and modified the results of R1-VL-7B built upon Qwen2.5-VL-7B. We will also make adjustments in the revised version based on your suggestions.
>
>
>
> > **Comment 2 + Question 1: Motivation** - "how performance differs when the number of rollouts increases or decreases? ...", "how the Perceptual and Reasoning difficulty groups change throughout the training process ..."
>
> - **Performance difference on different rollout numbers.**
>
> To address your concerns, we conduct a comparative experiment when rollout=2 and 32. The results are as follows:
>
> **Table 2: Performance difference (build upon DAPO) when rollout number is 2.**
>
> |                 | MathVerse | **MathVision** | MathVista | WeMath | HallusionBench |
> | :-------------: | :-------: | :------------: | :-------: | :----: | :------------: |
> | DAPO(rollout=2) |   48.8    |      25.9      |   69.3    |  67.9  |      69.1      |
> | Ours(rollout=2) |   48.7    |      27.1      |   70.1    |  69.7  |      67.3      |
>
> **Table 3: Performance difference (build upon DAPO) when rollout number is 32.**
>
> |                  | MathVerse | **MathVision** | MathVista | WeMath | HallusionBench |
> | :--------------: | :-------: | :------------: | :-------: | :----: | :------------: |
> | DAPO(rollout=32) |   50.9    |      28.8      |   70.0    |  69.8  |      70.6      |
> | Ours(rollout=32) |   51.9    |      28.9      |   72.2    |  71.0  |      71.3      |
>
> We also compare the results of rollout=8 with rollout=2 and 32:
>
> **Table 4: Performance comparison (build upon DAPO) between different rollout numbers.**
>
> |                  | MathVerse | **MathVision** | MathVista | WeMath | HallusionBench |
> | :--------------: | :-------: | :------------: | :-------: | :----: | :------------: |
> | Ours(rollout=2)  |   48.7    |      27.1      |   70.1    |  69.7  |      67.3      |
> | Ours(rollout=32) |   51.9    |      28.9      |   72.2    |  71.0  |      71.3      |
> | Ours(rollout=8)  |   51.9    |      29.0      |   72.2    |  71.8  |      71.4      |

---

> ### Author Response · Authors · 2025-11-26
> **Response to Reviewer jCT4 ---- Part 2/4**
>
> We can observe that:
>
> **(1) Rollout = 2 degenerates into PPO-like training.**
> With only one positive and one negative sample can participate in gradient updating, this setting naturally avoids extreme samples, but at the cost of completely losing GRPO’s benefits. Consequently, performance drops markedly across all benchmarks.
>
> **(2) Rollout = 32 does not alleviate variance collapse.**
> Although larger rollout numbers provide more samples, it seems they do not mitigate the normalization issue. Since its performance is consistent with or even slightly lower than that of rollout=8. We think that this might be due to the overfitting of the model.
>
> To better examine the situation during the training process of rollout=32, we performed a detailed step-by-step analysis of reward statistics across 60 training steps with 512 samples and their 32 rollout rewards. The empirical evidence clearly shows that variance collapse still happens though the rollout number increases. This is not an occasional event but a persistent and systemic phenomenon.
>
> **Table 5: The statistics of rewards about extreme samples across 60 training steps when rollout=32.**
>
> |                Training steps                 |   1   |  10   |  20   |  30   |  40   |  50   |  60   |
> | :-------------------------------------------: | :---: | :---: | :---: | :---: | :---: | :---: | :---: |
> | Effective samples (participating in training) |  414  |  402  |  424  |  394  |  383  |  369  |  355  |
> | Extreme success (31,30,29 correct/32 rollout) |  46   |  43   |  48   |  88   |  105  |  138  |  149  |
> |  Extreme failure (31,30,29 wrong/32 rollout)  |  117  |  90   |  99   |  72   |  65   |  31   |  42   |
> |             Total  Extreme Ratio              | 39.4% | 33.1% | 34.7% | 40.6% | 44.4% | 45.8% | 53.8% |
>
> We first exclude the samples with 32 identical rewards (i.e., variance = 0) for not participating in gradient updates. Among the remaining effective samples, **33%–53% samples exhibit the extreme reward patterns** (i.e., 29/30/31 correct or wrong in 32 rewards), which produce extremely small variance. That is, the extreme std samples are still widespread under the situation of rollout=32. Besides, the occurrence of this situation will increase as training deepens.
>
> - **How the Perceptual and Reasoning difficulty groups change throughout the training process.**
>
> To further illustrate how perceptual and reasoning difficulty groups evolve during training, we analyze the **standard deviation of each difficulty group** over 60 training steps. At every step, we computed the group-level reward std for the **3 perceptual difficulty groups** (low/mid/high entropy) and **10 reasoning difficulty groups** (different model confidence).
>
> **Table 6: The std changes of perceptual difficulty groups during 60 training steps.**
>
> | Training steps |   1    |   10   |   20   |   30   |   40   |   50   |   60   |
> | :------------: | :----: | :----: | :----: | :----: | :----: | :----: | :----: |
> |  Low entropy   | 0.4983 | 0.4352 | 0.4564 | 0.4638 | 0.4741 | 0.4899 | 0.4953 |
> | Medium-entropy | 0.4802 | 0.4494 | 0.4590 | 04474  | 0.4734 | 0.4778 | 0.4648 |
> |  High-entropy  | 0.4728 | 0.4695 | 0.4612 | 0.4501 | 0.4590 | 0.4750 | 0.4830 |
>
> **Table 7: The std changes of reasoning difficulty groups during 60 training steps.**
>
> |       Training steps       |   1    |   10   |   20   |   30   |   40   |   50   |   60   |
> | :------------------------: | :----: | :----: | :----: | :----: | :----: | :----: | :----: |
> |  group 1 (low confidence)  | 0.3741 | 0.3537 | 0.3307 | 0.2740 | 0.3720 | 0.4032 | 0.4288 |
> |          group 2           | 0.4007 | 0.3517 | 0.3705 | 0.4043 | 0.4096 | 0.4372 | 0.4007 |
> |          group 3           | 0.4163 | 0.3913 | 0.4113 | 0.3362 | 0.4043 | 0.4242 | 0.4620 |
> |          group 4           | 0.4287 | 0.4195 | 0.3873 | 0.4026 | 0.4358 | 0.4287 | 0.4659 |
> |          group 5           | 0.4610 | 0.4226 | 0.4567 | 0.4226 | 0.4739 | 0.4794 | 0.4835 |
> |          group 6           | 0.4841 | 0.4475 | 0.4650 | 0.4669 | 0.4835 | 0.4959 | 0.4871 |
> |          group 7           | 0.4882 | 0.4687 | 0.4822 | 0.4893 | 0.4589 | 0.5000 | 0.4946 |
> |          group 8           | 0.4938 | 0.4877 | 0.4942 | 0.4860 | 0.4986 | 0.4871 | 0.4994 |
> |          group 9           | 0.4705 | 0.4908 | 0.4930 | 0.4938 | 0.4978 | 0.4968 | 0.4854 |
> | group 10 (high confidence) | 0.4795 | 0.4990 | 0.4999 | 0.4996 | 0.4979 | 0.4967 | 0.4892 |
>
> We can observe that, whether in the perceptual or reasoning difficulty groups, the std within each group can remain stable continuously during the training process.
>
> Thanks again for your feedback! If you have any additional questions, please feel free to contact us!

---

> ### Author Response · Authors · 2025-11-26
> **Response to Reviewer jCT4 ---- Part 3/4**
>
> > **Comment 4 + Question 2 + Question 3: Experimental details** - "The values of the three $\alpha$-weights ...", "how the training dynamics behave ...", "Would performance continue to increase if the dataset size were scaled to over 10K samples? ..."
>
> - **The weighting coefficients**
>
>  We experimented with different combinations of three coefficients, and the results are presented in Table 8.
>
> **Table 8: The effects of different weighting coefficients of our method (build upon DAPO) on 5 benchmarks.**
>
> | $\alpha_{Ori}$ | $\alpha_{Percep}$ | $\alpha_{Reason}$ | MathVerse | **MathVision** | MathVista | WeMath | HallusionBench |
> | :------------: | :---------------: | :---------------: | :-------: | :------------: | :-------: | :----: | :------------: |
> |      0.1       |        0.2        |        0.7        |   50.7    |      28.6      |   71.6    |  71.8  |      71.2      |
> |      0.15      |       0.25        |        0.6        |   50.8    |      29.0      |   71.6    |  70.6  |      70.8      |
> |      0.2       |        0.1        |        0.7        |   51.2    |      28.4      |   71.5    |  70.4  |      71.0      |
> |      0.3       |        0.1        |        0.6        |   51.7    |      27.8      |   70.7    |  70.6  |      70.8      |
> |      0.4       |        0.3        |        0.3        |   51.9    |      27.9      |   71.4    |  70.5  |      71.1      |
> |      0.6       |        0.2        |        0.2        |   50.4    |      28.8      |   72.2    |  71.0  |      71.4      |
> |      0.7       |        0.1        |        0.2        |   51.1    |      28.3      |   70.3    |  71.1  |      69.8      |
>
> We can observe that while the performance on different benchmarks varies slightly with different settings, the method is relatively stable across a wide range of settings, with no significant degradation in results, indicating that our model is not overly sensitive to the specific choice of hyperparameters. This suggests that our method does not require extremely fine-tuned hyperparameters to perform effectively. Besides, we find that, usually a little $\alpha_{Ori}$ and relatively larger $\alpha_{Percep}$ and $\alpha_{Reason}$ are more likely to achieve a better effect, which can be used as a guidance in the application of our method.
>
> - **Training dynamics**
>
> Thank you for raising this important point! To thoroughly address your concerns regarding training dynamics, we will add detailed analyses in the `Appendix` of our revised version. Specifically, we include visualizations of **reward trajectories** and **advantage fluctuations** over the course of training, and directly compare them with the original baselines.
>
> We sincerely appreciate your suggestion, which helped us strengthen the clarity and completeness of our paper.
>
> - **Training on larger dataset**
>
> To further assess the scalability of our approach, we trained our method on the **ViRL39K dataset**[4], which is much larger in scale, covering comprehensive topics and categories, and reasoning with charts, diagrams, tables, documents, and spatial relationships. The results show that when trained on this larger dataset, our method achieves substantial improvements in performance, highlighting both the **effectiveness** and **scalability** of our approach.
>
> **Table 9: The performance of our method trained on ViRL39K (build upon DAPO) on 5 benchmarks.**
>
> |                  | MathVerse | **MathVision** | MathVista | WeMath | HallusionBench |
> | :--------------: | :-------: | :------------: | :-------: | :----: | :------------: |
> | DAPO(Geometry3K) |   50.4    |      27.6      |   70.7    |  69.4  |      68.6      |
> | Ours(Geometry3K) |   51.9    |      29.0      |   72.2    |  71.8  |      71.4      |
> |  Ours(ViRL39K )  |   52.4    |      29.9      |   73.8    |  72.0  |      72.5      |

---

> ### Author Response · Authors · 2025-11-26
> **Response to Reviewer jCT4 ---- Part 4/4**
>
> > **Question 4: The bias of reasoning difficulty groups** - " when the number of groups exceeds 40..., could this introduce statistical bias? ..."
>
> In our experimental setting, each batch contains 512 samples, thus, even with 48 groups, each group still includes approximately 10 samples and their associated 8 rollouts.
>
> To further examine whether the improvement at 48 groups on HallusionBench could be attributed to enhanced self-confidence rather than genuine reasoning gains, we extended our analysis by increasing the number of reasoning-difficulty groups to 64, 80, and 96.
>
> **Table 10: The comparison performance of different groups on HallusionBench.**
>
> |  48  |  64  |  96  |
> | :--: | :--: | :--: |
> | 71.0 | 69.8 | 69.1 |
>
> We can observe that, as the number of groups increases beyond 48, there is a noticeable decline in performance. This indicates that 48 is an optimal grouping size, rather than reflecting overconfidence or overfitting.
>
>
>
> [1] Yaowei Zheng, et al. EasyR1: An Efficient, Scalable, Multi-Modality RL Training Framework.
>
> [2] Xiangyan Liu, et al. NoisyRollout: Reinforcing Visual Reasoning with Data Augmentation. In NeurIPS 2025.
>
> [3] Jingyi Zhang, et al. R1-VL: Learning to Reason with Multimodal Large Language Models via Step-wise Group Relative Policy Optimization. In ICCV 2025.
>
> [4] Zhenhailong Wang, et al. Perception-Aware Policy Optimization for Multimodal Reasoning.

---

### Official Review · Reviewer_K5qY · 2025-10-31

**Soundness:** 2
**Presentation:** 2
**Contribution:** 2
**Rating:** 2
**Confidence:** 4

**Summary:**

This paper proposes difficulty-aware group normalization, a method to enhance the stability and effectiveness of reinforcement learning for multimodal large language models (MLLMs). The authors first identify that std-based reward normalization in GRPO is highly sensitive to extreme samples, causing instability in multimodal reasoning tasks. To address this, they introduce a difficulty-based re-grouping strategy that characterizes each sample’s difficulty through perceptual complexity and reasoning uncertainty. Samples are re-grouped by difficulty levels, and normalization is shared within each group to reduce sensitivity to outliers while preserving intra-group distinctions. The authors conduct experiments on multimodal math and hallucination benchmarks to validate the proposed method.

**Strengths:**

- The paper is overall clear and easy to follow.
- To the reviewer's knowledge, the proposed difficulty-based regrouping method is novel and conceptually straightforward.

**Weaknesses:**

- The definition of perceptual difficulty is not well validated. Why are images with higher entropy of eigenvalues of their image features considered perceptually harder? Moreover, why is perceptual difficulty assumed to be LLM-agnostic?

  Similarly, for reasoning difficulty, why is entropy used instead of a more direct measure such as average accuracy?

- Lack sufficient ablation analysis.

  - How sensitive are the results to the weighting coefficients of different normalized advantages?
  - What is the effect of group size on performance and stability?

- The proposed method combined with GRPO does not appear to improve over NoisyRollout when trained on the same amount of data. It remains unclear how the proposed approach achieves effectiveness compared to other baseline methods.

- Minor clarity issues:  There are two "+ours" in Table 2. It would be clearer if the authors explicitly distinguished their differences.

**Questions:**

Please refer to weaknesses sections for details.

**Details Of Ethics Concerns:**

Not much.

---

> ### Author Response · Authors · 2025-11-26
> **Response to Reviewer K5qY ---- Part 1/3**
>
> **Response to Reviewer $\color{green}{\text{K5qY}}$**
>
> We highly appreciate your insightful comments, your constructive criticism is invaluable in refining our work.  We notice that your comments and questions are mostly focused on our **experiments** and **the definition of our method**. For clear demonstration, we provide the point-to-point clarification and explanation below. If you have additional questions, we would be pleased to discuss them with you.
>
> > **Comment 1: The definition of perceptual and reasoning difficulty** - "Why are imgaes with higher entropy ... ? Why is perceptual difficulty assumed to be LLM-agnostic? Why is entropy used instead of ... average accuracy? "
>
> - **Why is perceptual difficulty assumed to be LLM-agnostic?**
>
> We apologize for any confusion caused. In fact, we did not intend to emphasize that perceptual difficulty is LLM-agnostic.
>
> The **perception process** (Multi-modal Large Language Model (MLLM) **transforms a given image into textual descriptions)** has the following two stages:
>
> 1. **Image $\to$ Embeddings**: the visual encoder first converts the image into a feature embedding that the LLM can understand.
> 2. **Embeddings $\to$ tokens**: the LLM leverages this visual embedding (same as textual embeddings) to perform logical reasoning.
>
> Over the past generation of MLLMs—from early LLaVA to Qwen-VL sequences and even recent Think-with-Image paradigm (e.g., DeepEyes)—research advances[1,2] in *perception* have almost exclusively focused on the **first stage**:
>
> - Higher-resolution or dynamic-resolution,
> - Improved patch extraction,
> - Enhanced visual feature utilization,
> - or Tools to enlarge/focus local visual details.
>
> These works consistently show that **the primary bottleneck in MLLM perception lies in the visual encoder.**Hence, in our paper, when we refer to “perceptual difficulty,” we specifically target the **difficulty for the visual encoder to extract useful representations**, rather than the second stage. Our analysis and grouping about perceptual difficulty therefore operate at the visual-embedding level, where perceptual bottlenecks originate.
>
> - **Why are images with higher entropy of eigenvalues of their image features considered perceptually harder?**
>
> **Perceptual difficulty** in our framework is defined based on **the complexity of visual embeddings**, which we quantify using spectral analysis of image patch covariances. Specifically, the entropy of the eigenvalue distribution from the covariance matrix reflects the amount of variance across spatial features in the image. Researchers in prior works[3,4,5] support that: high entropy indicates a more diverse distribution of visual features, implying a richer and more complex visual structure. This complexity makes it more challenging for visual model to recognize, and thus we associate higher entropy with greater perceptual difficulty.
>
> Besides, we conduct **empirical experiments** to validate whether our definition of visual entropy can represent perceptual difficulty. Specifically, during the evaluation stage, we conduct an analysis focusing on the **questions that the model answered incorrectly** on two benchmarks. We then use Gemini2.5 Pro to classify the sources of errors, extracting the samples with errors caused by the perception process.
>
> We want to examine whether these perception error samples are concentrated in the higher visual entropy groups as we defined. The intuition behind this approach is that samples belonging to higher-difficulty groups should naturally be harder for the model to perceive. Consequently, we expect these samples to exhibit higher error rates.
>
> **Table 1: The error rate of perception difficulty groups in perception errors on two benchmarks**.
>
> |                | low-entropy | medium-entropy | high-entropy |
> | :------------: | :---------: | :------------: | :----------: |
> |     Wemath     |    23.6%    |     31.3%      |    45.1%     |
> | HallusionBench |    21.2%    |     29.6%      |    49.2%     |
>
> We group the samples based on visual entropy, and calculate the proportion of incorrect answers in each group relative to the total number of perception errors.
>
> As shown in **Table 1**, the results can align with our expectations: images in the low-entropy group exhibit a significantly lower perceptual error rate compared to those in the high-entropy group. Our empirical findings are consistent with this intuition, further supporting the validity of our difficulty metrics.
>
> We appreciate your feedback on this point, and we will provide a clearer explanation of this in the revised version of our paper.

---

> ### Author Response · Authors · 2025-11-26
> **Response to Reviewer K5qY ---- Part 2/3**
>
> - **Why is entropy used for reasoning difficulty instead of a more direct measure such as average accuracy?**
>
> Thank you for your insightful question. In fact, during the early iterations of our method, we did experiment with using average accuracy as a proxy for reasoning difficulty. Specifically, we classified the difficulty of tasks into easy, medium, and hard categories based on the average accuracy from multiple rollouts of the model. This approach, combined with visual entropy as a proxy for perceptual difficulty, yielded the following results:
>
> **Table 2: The results of Average Accuracy + Visual Entropy (build upon GRPO) on 5 benchmarks.**
>
> |  | MathVerse | MathVision | MathVista | wemath | HallusionBench |
> | :-------------------------------: | :-------: | :--------: | :-------: | :----: | :------------: |
> |  GRPO  |   49.6    |    26.8    |   70.2    |  68.2  |      69.8      |
> | Average Accuracy + Visual Entropy |   51.5    |    28.0    |   71.4    |  70.6  |      70.5      |
>
> While this combined approach did show some performance improvement over the vanilla GRPO, it was not as competitive when compared to other baselines. Upon further analysis, we found that the reason for this outcome is that **accuracy (or average accuracy)** is the result of both perceptual and reasoning difficulties. It fails to isolate the specific challenges in the reasoning process, as it aggregates the effect of both perception and reasoning into a single metric. As a result, it does not provide a clear measure of the **difficulty in the reasoning process itself**.
>
> In contrast, sequence-level log probability directly captures the **uncertainty in the reasoning process**, providing a more targeted and nuanced measure of reasoning difficulty. This enables us to more effectively differentiate between tasks with varying levels of complexity and uncertainty in reasoning, which ultimately leads to more stable and interpretable model training.
>
> > **Comment 2: Ablation Study** - " How sensitive are the results to... ? What is the effect of ... ? "
>
> - **How sensitive are the results to the weighting coefficients of different normalized advantages?**
>
>  We experimented with different combinations of three coefficients, and the results are presented in Table 3.
>
> **Table 3: The effects of different weighting coefficients of our method (build upon DAPO) on 5 benchmarks.**
>
> | $\alpha_{Ori}$ | $\alpha_{Percep}$ | $\alpha_{Reason}$ | MathVerse | **MathVision** | MathVista | WeMath | HallusionBench |
> | :------------: | :---------------: | :---------------: | :-------: | :------------: | :-------: | :----: | :------------: |
> |      0.1   |        0.2        |        0.7        |   50.7    |      28.6      |   71.6    |  71.8  |      71.2      |
> |      0.15   |       0.25        |        0.6        |   50.8    |      29.0      |   71.6    |  70.6  |      70.8      |
> |      0.2     |        0.1        |        0.7        |   51.2    |      28.4      |   71.5    |  70.4  |      71.0      |
> |      0.3     |        0.1        |        0.6        |   51.7    |      27.8      |   70.7    |  70.6  |      70.8      |
> |      0.4      |        0.3        |        0.3        |   51.9    |      27.9      |   71.4    |  70.5  |      71.1      |
> |      0.6      |        0.2        |        0.2        |   50.4    |      28.8      |   72.2    |  71.0  |      71.4      |
> |      0.7      |        0.1        |        0.2        |   51.1    |      28.3      |   70.3    |  71.1  |      69.8      |
>
> We can observe that while the performance on different benchmarks varies slightly with different settings, the method is relatively stable across a wide range of settings, with no significant degradation in results, indicating that our model is not overly sensitive to the specific choice of hyperparameters. This suggests that our method does not require extremely fine-tuned hyperparameters to perform effectively. Besides, we find that, usually a little $\alpha_{Ori}$ and relatively lager $\alpha_{Percep}$ and $\alpha_{Reason}$ are more likely to achieve a better effect, which can be used as a guidance in the application of our method.
>
> - **What is the effect of group size on performance and stability?**
>
> We explored the impact of different group sizes on both performance and stability. The experimental results are as follows:
>
> **Table 4: The effects of group size (build upon DAPO) on 5 benchmarks.**
>
> | rollout | MathVerse | **MathVision** | MathVista | WeMath | HallusionBench |
> | :-----: | :-------: | :------------: | :-------: | :----: | :------------: |
> |    2  |   48.7    |      27.1   |   70.1    |  69.7  |  67.3  |
> |    4  |   50.1    |      28.4      |   71.5    |  70.2  |   68.9  |
> |    8  |   51.9    |      29.0      |   72.2    |  71.8  |   71.4  |
> |   16  |   52.1    |      29.2      |   72.1    |  71.2  |  71.2  |
> |   24  |   51.7    |      29.0      |   71.9    |  71.9  | 71.5     |
> |   32  |   51.9    |      28.9      |   72.2    |  71.0  |  71.3   |

---

> ### Author Response · Authors · 2025-11-26
> **Response to Reviewer K5qY ---- Part 3/3**
>
> We observed that when the group size is smaller than 8, the performance improves as the group size increases. Notably, when the number of rollouts is reduced to 2, the model reverts to PPO. As the group size continues to increase beyond 8, the improvement in performance becomes less pronounced, eventually stabilizing at a stable value.
>
> These results indicate that while increasing the number of rollouts can lead to better performance,  after a certain point, beyond which further increases in group size do not significantly contribute to performance improvement. This shows the importance of selecting an appropriate group size to balance computational cost and model performance.
>
>
>
> > **Comment 3: Performance compared with NoisyRollout and other baselines** - " does not appear to improve over NoisyRollout when ... . It remains unclear how the proposed approach achieves effectiveness ... ? "
>
> Thank you for your comment! We would like to clarify the following points regarding the comparison with NoisyRollout and other baseline methods.
>
> 1. **NoisyRollout** employs a simple yet effective **data augmentation method**, which mixes training trajectories from both clean and moderately distorted images[6]. In contrast, our approach does not introduce any additional data or data augmentation techniques. Instead, it improves the efficiency of utilizing the existing training data, which enables our method to achieve performance comparable to NoisyRollout, even with the same amount of training data.
> 2. **Regarding other baselines**, it is important to note that **they are typically trained on significantly larger datasets**. Despite this, our method demonstrates competitive performance, even when trained on smaller datasets. To further assess the scalability of our approach, we trained our method on the **ViRL39K dataset**[7], which is much larger in scale. The results show that when trained on this larger dataset, our method achieves substantial improvements in performance, surpassing several baselines trained on even larger datasets. This highlights both the **effectiveness** and **scalability** of our approach.
>
> **Table 5: The performance of our method trained on ViRL39K (build upon DAPO) on 5 benchmarks.**
>
> |                                        | MathVerse | **MathVision** | MathVista | WeMath | HallusionBench |
> | :------------------------------------: | :-------: | :------------: | :-------: | :----: | :------------: |
> |        NoisyRollout(Geometry3K)        |   53.2    |      28.5      |   72.6    |  69.6  |      72.1      |
> | Qwen2.5VL-7B-Instruct+DAPO(Geometry3K) |   50.4    |      27.6      |   70.7    |  69.4  |      68.6      |
> |            Ours(Geometry3K)            |   51.9    |      29.0      |   72.2    |  71.8  |      71.4      |
> |             Ours(ViRL39K )             |   52.4    |      29.9      |   73.8    |  72.0  |      72.5      |
>
> We hope this addresses your concerns and provides clearer insights into the effectiveness of our method when compared to both NoisyRollout and other baseline methods.
>
>
>
> > **Comment 4: There are two "+ours" in Table 2** - "It would be clearer if the authors explicitly distinguished their differences. "
>
> Thanks for your kind suggestions, and sorry for such confusion! The first "ours" represents the performance based on the baseline of GRPO, and the second "ours" represents the performance based on the baseline of DAPO. We will follow your suggestion and modify the clarification in our revised version.
>
>
>
>
>
> [1] Zhenhailong Wang, et al. Perception-Aware Policy Optimization for Multimodal Reasoning.
>
> [2] En Yu, et al. Perception-R1: Pioneering Perception Policy with Reinforcement Learning. In NeurIPS 2025.
>
> [3] Ruth Rosenholtz, et al. Measuring visual clutter. In JOV, August 2007, Vol.7, 17.
>
> [4] Mengdi Chu, et al. What Makes a Visualization Image Complex?
>
> [5] Norberto M. Grzywacz. Perceptual Complexity as Normalized Shannon Entropy. In Entropy 2025.
>
> [6] Xiangyan Liu, et al. NoisyRollout: Reinforcing Visual Reasoning with Data Augmentation. In NeurIPS 2025.

---

### Official Review · Reviewer_dxEx · 2025-11-04

**Soundness:** 2
**Presentation:** 3
**Contribution:** 2
**Rating:** 4
**Confidence:** 3

**Summary:**

The paper identifies a key limitation of GRPO when applied to multimodal reasoning tasks—the instability of standard-deviation-based normalization due to extreme reward outliers. Unlike text-only LLMs, multimodal models suffer amplified distortions since both perceptual and reasoning errors affect their outputs. To mitigate this, the authors introduce a difficulty-aware group normalization method. Each sample is first characterized by its perceptual complexity (via visual entropy) and reasoning uncertainty (via model confidence), then grouped by difficulty level. Standard deviation is shared within each difficulty group, preserving relative ranking within groups while stabilizing learning against outliers. Though, I appreciate the idea (especially behind the visual entropy), I think many claims need to validated.

**Strengths:**

1. The paper presents a well-motivated approach to improve GRPO stability by dividing samples into difficulty-based groups and computing group-specific normalization factors.
2. The use of image entropy to quantify perceptual difficulty is an insightful way to model visual reasoning complexity.

**Weaknesses:**

I appreciate the work. Here are my comments to improve further,

1. The paper’s core premise—that std-based group normalization is highly sensitive to extreme samples—lacks concrete empirical evidence. The authors should analyze when and how such cases occur, as infrequent occurrences might naturally average out during training, reducing the claimed impact of this issue.

2. The use of image entropy as a proxy for perceptual reasoning difficulty is not clearly justified. While entropy of eigenvalues after PCA may capture visual diversity or texture complexity, it does not directly indicate the model’s perceptual reasoning error. The observed difficulty might instead reflect dataset bias—i.e., the model having seen more or fewer examples of such images during training—rather than inherent perceptual complexity.

3. “This formulation reflects the model’s internal confidence: High and consistent L(Qs) indicates a reliable reasoning chain, whereas low or fluctuating L(Qs) reflects epistemic uncertainty, implying that more challenging reasoning sample.” — I think this motivation needs validation. How is a lower-confidence sample associated with a more challenging reasoning sample? This should be demonstrated through quantitative analysis or supported by theoretical justification (which may be difficult).

4. Could you please explain the idea behind the unified formulation for advantage function across difficulty groups, equations 12, 13, 18, 19?

5. Based on the motivation, the unified advantage function across reasoning types in equation 20, why do we still need the original advantage function?

6. Could you please provide statistical significance tests for table 1 to confirm that the improvement gains are meaningful especially on MathVision, MathVista, WeMath, and HallusionBench?

7. “Compared with those either distilled from large-scale chain-of-thought data or employing complex data augmentation strategies, our method, utilizing only 2.1k train- ing samples, achieves comparable or even superior performance, significantly demonstrating our effectiveness.” - Which component in the proposed method leads to this sample efficiency? Authors can elaborate on it for further understanding of this study.

**Questions:**

Please see above.

---

> ### Author Response · Authors · 2025-11-26
> **Response to Reviewer dxEx ---- Part 1/4**
>
> **Response to Reviewer $\color{red}{\text{dxEx}}$**
>
> We highly appreciate your invaluable comments, which can inspire us to greatly improve our paper! Below, we provide the point-to-point responses to address your concerns and clarify the confusion of our proposed method. If you have additional questions, we would be pleased to discuss them with you.
>
> > **Comment 1: Core premise and motivation**  - " ...the std-based group normalization is highly sensitive to extreme samples lacks concrete empirical evidence ...", " ... should analyze  when and how such cases occur... "
>
> Thank you for raising this point! We fully understand your concerns and it is indeed necessary to be well analyzed. We performed a detailed step-by-step analysis of reward statistics across 60 training steps with 512 samples and their 8 rollout rewards. The empirical evidence clearly shows that the existence of extreme samples is not an occasional event but a persistent and systemic phenomenon.
>
> **Table 1: The statistics of rewards about extreme samples within the batch across 60 training steps.**
>
> |                Training steps                 |   1   |  10   |  20   |  30   |  40   |   50   |  60   |
> | :-------------------------------------------: | :---: | :---: | :---: | :---: | :---: | :----: | :---: |
> | Effective samples (participating in training) |  323  |  327  |  324  |  322  |  297  |  314   |  306  |
> |     Extreme success (7 correct & 1 wrong)     |  41   |  39   |  48   |  66   |  78   |   60   |  82   |
> |     Extreme failure (7 wrong & 1 correct)     |  78   |  89   |  74   |  51   |  54   |   54   |  51   |
> |             Total  Extreme Ratio              | 36.8% | 39.1% | 37.7% | 36.3% | 44.4% | 36.31% | 43.5% |
>
> First, groups with 8 identical rewards (i.e., variance = 0) constitute 35%–46% of all samples at every training step. We first exclude these groups for not participating in gradient updates.
>
> Second, during the training process, there are **31%–44% samples exhibit the 7:1 extreme reward patterns** (i.e., 7/8 correct or wrong) among the remaining effective samples, which produce extremely small variance. Besides, the occurrence of this situation will increase as training deepens.
>
> These findings demonstrate that **the instability of std-based normalization is structural rather than incidental:** multimodal reasoning tasks naturally contain a large proportion of very easy and very hard samples, leading to unstable and unreliable advantage scaling. This directly motivates our difficulty-aware regrouping strategy, which stabilizes normalization by ensuring that variance is computed only within samples of comparable difficulty.
>
> Thanks again for pointing it out! We will carefully revise our paper, taking your feedbacks into account!
>
>
>
> > **Comment 2: Image entropy as a proxy for perceptual difficulty**  - " ... entropy of eigenvalues after PCA may capture visual diversity or texture complexity, it does not directly indicate the model's perceptual reasoning error... The observed difficulty might instead reflect dataset bias... "
>
> We are very grateful to the reviewer for this deep and crucial question! We acknowledge that the visual encoder (VE), shaped by its pre-training and SFT process, **carries inherent biases**. This raises a concern about whether the visual entropy metric captures "inherent complexity" or simply "reflects this bias".
>
> We concede that no feature representation from a pre-trained model can be truly "objective." The VE from a model like Qwen2.5-VL has its own "worldview" shaped by its training data. Actually, our image entropy metric is designed to quantify the perceptual difficulty **from the perspective of the base model itself**, rather than **some absolute, objective complexity**. From this model-centric viewpoint, the entropy of PCA eigenvalues on patch features becomes a highly effective proxy.
>
> We acknowledge that high entropy might reflect dataset bias (the VE has seen fewer similar images) rather than inherent complexity. We argue that for the purpose of our RL optimization, **these two sources of difficulty are functionally equivalent and equally important to identify.**
>
> - Case A (Inherent Complexity): An image is dense with information (e.g., intricate textures, multiple occluded objects).
> - Case B (Data Bias): An image is simple but belongs to a domain underrepresented in the VE's pre-training data (e.g., a specific type of medical scan or abstract art).
>
> Both cases signal that the perceptual grounding is challenging and needs to be treated carefully.

---

> ### Author Response · Authors · 2025-11-26
> **Response to Reviewer dxEx ---- Part 2/4**
>
> Besides, we conduct **empirical experiments** to validate whether our definition of visual entropy can represent perceptual difficulty. Specifically, during the evaluation stage, we conduct an analysis focusing on the **questions that the model answered incorrectly** on two benchmarks. We then use Gemini2.5 Pro to classify the sources of errors, extracting the samples with errors caused by the perception process.
>
> We want to examine whether these perception error samples are concentrated in the higher visual entropy groups as we defined. The intuition behind this approach is that samples belonging to higher-difficulty groups should naturally be harder for the model to perceive. Consequently, we expect these samples to exhibit higher error rates.
>
> **Table 2: The error rate of perception difficulty groups in perception errors on two benchmarks**.
>
> | | low-entropy | medium-entropy | high-entropy |
> | :------------: | :---------: | :------------: | :----------: |
> |     Wemath     |    23.6%    |   31.3%   |    45.1%     |
> | HallusionBench |    21.2%    |    29.6%   |    49.2%     |
>
> We group the samples based on visual entropy, and calculate the proportion of incorrect answers in each group relative to the total number of perception errors.
>
> As shown in **Table 2**, the results can align with our expectations: images in the low-entropy group exhibit a significantly lower perceptual error rate compared to those in the high-entropy group. Our empirical findings are consistent with this intuition, further supporting the validity of our perceptual difficulty metrics.
>
>
>
> > **Comment 3: Model Confidence**  - " ... this motivation needs validation. How a lower-confidence sample is associated with a more challenging reasoning sample... "
>
> - **Why model confidence naturally serves as a proxy for reasoning difficulty/uncertainty**
>
> Researchers in [1,2] propose that "one measure of uncertainty is the predictive entropy of the output distribution, which measures the information one has about the output given the input[3]. The predictive entropy for an input sentence $\bf{x}$ is the conditional entropy ($H$) of the output random variable ($Y$) with realization $y$ given $\bf{x}$."
> $$
> {\rm{PE}}({\bf{x}})=H(Y| {\bf{x}})=-\sum _{y}P(\,y| {\bf{x}})\mathrm{ln}P(\,y| {\bf{x}}).
> $$
> Researchers[4] also hypothesize that when a model knows the answer to a particular question, it is confident in its response, and would result in an answer distribution with small entropy. Conversely, when a model is unsure about its response, it will lead to an answer distribution with high entropy, thus implying a more challenging reasoning process.
>
> This aligns directly with our formulation: the **sequence-level log probabilities** we compute is theoretically linked to the notion of *semantic entropy* and represent the joint likelihood of the entire reasoning chain. A low log-probability corresponds to a flat or high-entropy output distribution, reflecting uncertainty in the reasoning trajectory, while a high log-probability corresponds to a confident, low-entropy distribution.
>
> Besides, we also conduct empirical experiments to validate this motivation.
>
> Specifically, during the evaluation stage, we conduct an analysis focusing on the **questions that the model answered incorrectly** on two benchmarks. We then use Gemini2.5 Pro to classify the sources of errors, extracting the samples with errors caused by the reasoning process.
>
> We want to examine whether these reasoning error samples are concentrated in the lower model confidence groups as we defined. The intuition behind this approach is that samples belonging to lower confidence groups represent the uncertainty during the reasoning process of the model, thus demonstrating potential reasoning failures. Consequently, we expect these samples to exhibit higher error rates.
>
> **Table 3: The error rate of reasoning difficulty groups in reasoning errors on two benchmarks**.
>
> |  | group 1 (low confidence) | group 2 | group 3 | group 4 | group 5 | group 6 | group 7 | group 8 | group 9 | group 10 (high confidence) |
> | :------------: | :----------------------: | :-----: | :-----: | :-----: | :-----: | :-----: | :-----: | :-----: | :-----: | :------------------------: |
> |     Wemath     |          13.4%           |  12.6%  |  13.2%  |  12.7%  |  11.6%  |  9.1%   |  9.7%   |  7.2%   |  6.7%   |            4.2%            |
> | HallusionBench |          12.7%           |  11.9%  |  11.2%  |  11.7%  |  10.0%  |  9.8%   |  9.0%   |  8.2%   |  8.3%   |            7.1%            |
>
> We group the samples based on model confidence, and calculate the proportion of incorrect answers in each group relative to the total number of reasoning errors.
>
> As shown in **Table 3**, the results can align with our expectations: samples with **lower model confidence** correspond to **higher reasoning error rates**. Our empirical findings are consistent with this intuition, further supporting the validity of our difficulty metrics.

---

> ### Author Response · Authors · 2025-11-26
> **Response to Reviewer dxEx ---- Part 3/4**
>
> > **Comment 4 + Comment 5 + Comment 7: Detailed explanation of the method and experiment**  - "The idea behind equations 12, 13, 18, 19... ", "... why do we still need the original advantage function?", "Which component in the proposed method leads to this sample efficiency? ..."
>
> - **The idea behind equations 12, 13, 18, 19.**
>
> Equations 12, 13, 18, and 19 are the cores of our method. The primary idea behind the advantage function is to normalize rewards within groups of similar difficulty, so that the variance used for advantage scaling reflects the appropriate difficulty level rather than being dominated by extreme-easy or extreme-hard samples.
>
> Specially,
>
> (1) All four equations follow the GRPO normalization template:
> $$
> \begin{align}
> \hat A_{i}
> = \frac{r_i - mean(\{ r_1, r_2, \dots, r_G\})}{std(\mathcal{R})}.
> \end{align}
> $$
> The numerator keeps **intra-sample distinctions** among rollouts for the same input (as in GRPO). The denominator replaces the unstable per-sample std with a **group-level std**, computed over samples of comparable difficulty.
>
> (2) **Eqs. 12 & 13**: Perceptual difficulty–based normalization
>
> These equations assign each sample to a perceptual group (low/medium/high image entropy), computing shared normalization scale per perceptual group $std(\mathcal{R}_a)$ and advantages normalized according to perceptual difficulty $A^{\text{Perceptual}}$.
>
> (2) **Eqs. 18 & 19**: Reasoning difficulty–based normalization
>
> These equations reuse exactly the same structure of Eqs. 12 & 13, but the grouping is based on model confidence. Thus, we compute shared scale for samples with similar model confidence $std(\mathcal{R}_u)$ and advantages normalized according to reasoning difficulty $A^{\text{Reasoning}}$.
>
> The consistent mathematical structure makes the two normalized advantages directly comparable and can be combined with the original GRPO advantage.
>
> - **Why do we still need the original advantage function?**
>
> The original GRPO advantage is retained in Eq. 20 because the perceptual and reasoning-based difficulty advantages are **auxiliary normalization signals**, not replacements for the core GRPO mechanism. While difficulty-aware normalization stabilizes the variance estimation for extreme samples, the original GRPO advantage preserves the **relative reward differences within each rollout group**, which is essential for effective policy improvement. Removing it would discard the primary optimization signal that GRPO is built upon, whereas combining all three terms enables us to stabilize normalization.
>
> For further validation of this opinion, we conduct empirical experiments on removing the original advantage term. The results are as follows:
>
> **Table 4: Performance comparisons about the original advantage term (build upon DAPO) over 5 benchmarks.**
>
> |                                  | MathVerse | MathVision | MathVista | wemath | HallusionBench |
> | :------------------------------: | :-------: | :--------: | :-------: | :----: | :------------: |
> |               DAPO               |   50.4    |    27.6    |   70.7    |  69.4  |      68.6      |
> | Our(w/o original advantage term) |   51.5    |    28.0    |   71.4    |  70.5  |      69.5      |
> |               Ours               |   51.9    |    29.0    |   72.2    |  71.8  |      71.4      |
>
> We can find that compared with DAPO, the results of removing the original advantage term have a certain improvement on benchmarks, but the overall effect is not as good as retaining this term.
>
> - **Which component in the proposed method leads to this sample efficiency?**
>
> Thank you for pointing out this aspect. The sample efficiency of our method primarily comes from the **difficulty-aware normalization mechanism**, which improves optimization quality rather than relying on **more data or stronger supervision**.
>
> **Difficulty-aware regrouping reduces gradient noise and maximizes the value of each training sample.** In standard GRPO, when the rollout rewards collapse to nearly all 0 or all 1, the per-group standard deviation becomes extremely small, leading to unstable advantage scaling, disproportionately large gradients from extreme samples, and inefficient use of limited data. Our difficulty-based regrouping ensures that **each sample is normalized with others of comparable perceptual or reasoning difficulty**, preventing variance collapse and dramatically **reducing gradient noise**. This means each data point contributes a more reliable and informative gradient, enabling the model to learn effectively.
>
> In summary, the key to our sample efficiency is that **difficulty-aware normalization improves the quality of each gradient step**, enabling GRPO-style reasoning optimization to make substantially better use of each training sample—even when only 2.1k samples are available.

---

> ### Author Response · Authors · 2025-11-26
> **Response to Reviewer dxEx ---- Part 4/4**
>
> > **Comment 6: Statistical significant tests for Table 1 of the paper**  - "Could you please provide ... to confirm the improvement gains are meaningful ... "
>
> Of course! We fully understand your concerns that the reported improvements are not just random fluctuations but are statistically meaningful.
>
> To address this, we re-run our experiments of our proposed method using 3 random seeds for data sampling during training on the Geometry3K dataset, and report the **mean and standard deviation** of the performance across these runs. The following results demonstrate the effectiveness of our method.
>
> **Table 5: Statistical significant tests for Table 1 of the paper.**
>
> |            Model             | MathVerse | MathVision | MathVista |  WeMath  | HallusionBench |
> | :--------------------------: | :-------: | :--------: | :-------: | :------: | :------------: |
> |             DAPO             |   50.4    |    27.6    |   70.7    |   69.4   |      68.6      |
> | DAPO + Perceptual regrouping |   52.3    |    28.0    |   71.4    |   70.9   |      70.9      |
> | DAPO + Reasoning regrouping  |   52.3    |    28.3    |   71.6    |   70.8   |      68.9      |
> |         DAPO + Both          | 51.9±0.3  |  29.0±0.4  | 72.1±0.6  | 71.6±0.4 |    71.0±0.3    |
>
>
>
> [1] Sebastian Farquhar, et al. Detecting hallucinations in large language models using semantic entropy. In Nature 2024
>
> [2] Dang Nguyen, et al. Beyond Semantic Entropy: Boosting LLM Uncertainty Quantification with Pairwise Semantic Similarity. In ACL 2025.
>
> [3] Lindley, D. V. On a measure of the information provided by an experiment. In Ann. Math. Stat 1956.
>
> [4] Saurav Kadavath, et al. Language Models (Mostly) Know What They Know.

---

### Author Response · Authors · 2025-12-04

Dear Area Chair and Reviewers,

We truly appreciate your efforts and valuable suggestions in reviewing our paper. We are glad that most reviewers reached a positive consensus on our work's motivation, presentation, novelty, and experimental effectiveness. Since we received a decent number of reviews, and for your convenience, we provide a summary on the reviewers’ major feedback and our corresponding actions:

> **Strengths**

- **Motivation**:
  - "The paper presents a well-motivated approach to improve GRPO stability."  (Reviewer $\color{red}{\text{dxEx}}$)
- **Novelty**:
  - "The proposed difficulty-based regrouping method is novel and conceptually straightforward." (Reviewer $\color{green}{\text{K5qY}}$)
  - "This algorithm demonstrates novelty in addressing multimodal-specific challenges." (Reviewer $\color{blue}{\text{Qc3t}}$)
- **Presentation**:
  - "The paper is overall clear and easy to follow."(Reviewer $\color{green}{\text{K5qY}}$)
- **Promising results**:
  - "It attains comparable or superior performance to prior work while using only 2K training samples." (Reviewer $\color{orange}{\text{jCT4}}$)
  - "It achieves accuracy improvements on MathVista, MathVision, and WeMath benchmarks." (Reviewer $\color{orange}{\text{jCT4}}$)

> **Responses**

- **Reviewer  $\color{red}{\text{dxEx}}$**:
  - Motivation Validation1: We explain **the existence of extreme samples during training,** and std-based group normalization is highly sensitive to these extreme samples (`Appendix D`).
  - Motivation Validation2: We verify the feasibility of **utilizing image entropy as a proxy for perceptual difficulty** and **model confidence as a proxy for reasoning difficulty** from the perspective of theoretical and empirical analysis (`Appendix E`).
  - Detailed explanation: We explain the idea behind equations, original advantage function in our method .
  - More experimental evaluation: We offer statistical significant tests for Table 1 of the paper.
- **Reviewer $\color{green}{\text{K5qY}}$**:
  - Further illustration: We further validate the feasibility of the definition of perceptual and reasoning difficulty.
  - Further hyperparameter analysis: We provide the analysis of different combinations of three coefficients (`Appendix F`) and the effects of group size (`Appendix G`).
  - More experimental evaluation: We further analysis the comparison with NoisyRollout and other baseline methods.
- **Reviewer $\color{orange}{\text{jCT4}}$**:
  - More experimental evaluation: We further analysis the comparison with Reinforce++ and Dr.GRPO
  - Further analysis: We provide the analysis of different combinations of three coefficients (`Appendix F`), the effects of group size (`Appendix G`), and how the perceptual and reasoning difficulty groups change throughout the training process.
- **Reviewer $\color{blue}{\text{Qc3t}}$**:
  - Further hyperparameter analysis: We provide the analysis of different combinations of three coefficients (`Appendix F`).
  - Two Metrics Validation: We conduct experiments to validate our difficulty metrics actually capture the intended concepts(`Appendix E`).

We are pleased that most of the concerns have been clarified, and reviewers expressed satisfaction with our paper and acknowledged our efforts. Reviewer $\color{blue}{\text{Qc3t}}$ **appreciate our efforts  for the detailed and thorough response, and keep supporting us**! It's a pity that we cannot continue to clarify the remaining concerns that might exist of dear reviewers. We sincerely hope that our rebuttal and the revised version **clearly convey the motivation, contributions, analysis, and improvements we have incorporated to dear AC**. Thank you very much for your time and consideration!

Once again, we deeply appreciate the time and expertise that dear Area Chair and Reviewers have shared with us. Your encouraging feedback motivates us to continue advancing this work for the broader community！


Best regards,

Authors of Paper 17401

---

### Meta-Review · Area_Chair_QUUi · 2025-12-26

**Summary:**

The proposed method introduces unnecessary complexity with multiple hyperparameters and weighting coefficients while failing to demonstrate significant performance gains over simpler baselines like NoisyRollout or alternative normalization strategies. The experimental results do not convincingly justify the complex grouping mechanism compared to straightforward solutions like increasing rollout counts or applying batch-level normalization. The fundamental premise regarding the severity of standard deviation instability caused by outliers remains inadequately supported to warrant the proposed architectural changes.

**Reviewer Concerns:**

The concern regarding the method's performance superiority over simpler data augmentation baselines like NoisyRollout remains outstanding, as the proposed approach yields mixed results on key benchmarks despite its added complexity, and the theoretical distinction between inherent perceptual difficulty and dataset bias remains partially conflated.

**Reviewer Scores:**

Reviewer dxEx would maintain the marginally negative score as the theoretical ambiguity between inherent perceptual complexity and dataset bias.
Reviewer jCT4 would retain the borderline rating since the marginal performance gains over the baseline do not convincingly justify the additional architectural complexity.
Reviewer Qc3t will maintain the score as explicitly confirmed in the post-rebuttal correspondence.
Reviewer K5qY would likely lower the rating as the comparative results confirmed the method fails to outperform the simpler NoisyRollout baseline on key benchmarks.

---

### Decision · Program_Chairs · 2026-01-26

Reject